# Transcriptomic analysis reveals that pyruvate kinase potentially plays a key role in the differentiation of *Spirometra mansoni* proglottids by regulating the glycolysis pathway

Ke Zhou[1], Cheng Yue Cao[2], Si Si Ru[1], Rui Jie Wang[1], Jie Hao[1], Xi Zhang[1]*

1 Department of Parasitology, School of Basic Medical Sciences, Zhengzhou University, Zhengzhou, Henan, China, 2 School of Life Sciences, Zhengzhou University, Zhengzhou, Henan, China

* zhangxi@zzu.edu.cn

## Abstract

### Background

The differentiation and maturation of proglottids constitute the basis for the growth and development of tapeworms. However, little is known about the molecular mechanisms underlying the differentiation of the proglottids of *Spirometra mansoni*.

### Methodology/Principal findings

Here, the nanopore sequencing method was used to perform full-length transcriptomic analysis of 3 types of proglottids (scolex-neck-immature proglottids, SNIPs; mature proglottids, MPs; and gravid proglottids, GPs) of *S. mansoni*. Comparative transcriptomic analysis revealed that pyruvate kinase (PK) is a key gene affecting segmental differentiation. The PK family members of *S. mansoni* (*Sm*PKs) were subsequently screened and systematically analysed. Moreover, a representative member, *Sm*PK1, was chosen for cloning, expression and functional characterization. A total of 4,486 differentially expressed genes (DEGs) were identified across the 3 proglottid types. GO analysis revealed that the DEGs were enriched mostly in metabolism-related terms. KEGG enrichment analysis and GSEA further revealed that the degree of enrichment of the glycolysis pathway gradually increased as the segments developed and matured. Protein–protein interaction (PPI) analysis confirmed that PK occupies a central position among energy metabolism-related genes and plays key roles in glycolysis. On the basis of the omics data, 4 *Sm*PKs were identified. Phylogenetic analysis revealed that *Sm*PKs have undergone varying degrees of evolution and exhibit high diversity. The optimal reaction conditions for recombinant *Sm*PK1 (r*Sm*PK1) were 37 °C and pH 8.0, and the addition of K⁺/Mg²⁺ significantly enhanced its catalytic activity. Tannic acid significantly inhibited the activity of *Sm*PK1 *in vitro*, reduced the production of pyruvate, and forced the

**Data availability statement:** All the data supporting the findings of this article are included in the main article and its additional files. The RNA-seq data from this study have been deposited in the NCBI Sequence Read Archive (SRA) under the BioProject accession number PRJNA1248218. The raw sequencing data are accessible through the SRA accession numbers SRR33040041 to SRR33040049.

**Funding:** This work was supported by Intergovernmental international scientific and technological innovation cooperation projects of National Key Research and Development Program (2024YFE0199100 to XZ), Fundamental Research Project of key scientific research in Henan Province (24ZX003 to XZ) and National Natural Science Foundation of China (81971956 to XZ). The funders had no role in study design, data collection and analysis, decision to publish, or preparation of the manuscript.

**Competing interests:** The authors have declared that no competing interests exist.

organism to compensate for the energy supply through rapid lipolysis and delayed glycogen depletion, thereby affecting energy metabolism in tapeworms.

## Conclusions

This study provides the first comprehensive characterization of gene expression profiles across different proglottids of *S. mansoni*. PK plays a pivotal role in proglottid differentiation, and this finding lays the foundation for further exploration of the differentiation mechanism of segments in tapeworms.

## Author summary

Sparganosis is a food/water-borne zoonotic parasitic disease caused by the plerocercoid larvae of *Spirometra* tapeworms. The differentiation and maturation of proglottids constitute the basis for the growth and development of tapeworms and are key to understanding their pathogenic characteristics in detail. However, little is known about the molecular mechanisms underlying the differentiation of the proglottids of *S. mansoni*. Here, the nanopore sequencing method was used to perform full-length transcriptomic analysis of 3 types of proglottids of *S. mansoni*. *Sm*PK was ultimately selected for further analysis. *Sm*PK1 functional characterization was performed through enzyme activity assays and *in vitro* inhibition experiments. These results suggested that a total of 4,486 differentially expressed genes (DEGs) were identified across the 3 proglottid types. Comparative transcriptomic analysis revealed that PK occupies a central position among energy metabolism-related genes and its expression levels gradually increased as the segments developed and matured. Further analysis of the *Sm*PK family identified four *Sm*PK members (*Sm*PK1–*Sm*PK4). Phylogenetic analysis revealed that *Sm*PKs exhibit high diversity and are scattered across different cestode groups. Enzyme activity assays confirmed that recombinant *Sm*PK1 exhibits native enzymatic activity and that tannic acid exerts dose-dependent inhibitory effects on r*Sm*PK1. Further *in vitro* inhibition experiments using tannic acid revealed that pyruvate kinase activity was continuously inhibited, accompanied by a consistent and synchronous decrease in pyruvate content, a time-dependent consumption of triglycerides, and a gradual depletion of total sugar reserves. These results suggest that pyruvate kinase may influence tapeworm proglottid differentiation by regulating the glycolytic pathway.

## Introduction

Sparganosis is a food/water-borne zoonotic parasitic disease caused by the plerocercoid larvae of *Spirometra* tapeworms [1]. To date, over 2,000 cases of sparganosis have been reported worldwide, with the majority predominantly documented in East Asian and Southeast Asian countries [2]. The main manifestation of sparganosis

is larval migration syndrome, which can cause multiple lesions, including in subcutaneous tissues, the central nervous system, oral and maxillofacial regions, visceral organs, and eyes, ultimately resulting in tissue damage, paralysis, blindness, and even death [3]. Currently, sparganosis is caused mainly by the species *Spirometra mansoni* [4]. As sparganosis is mostly sporadic and has not received sufficient attention for a long time, it has become a neglected parasitic disease. Fundamental biological research on *S. mansoni* remains limited; therefore, improving our understanding of the pathogenic characteristics of *S. mansoni* is essential.

The differentiation and maturation of proglottids constitute the basis for the growth and development of tapeworms and are key to understanding their pathogenic characteristics in detail. Adult *S. mansoni* show characteristic body differentiation into the scolex, neck and strobila. The neck section is a very small part that occurs after the scolex section, and its function is to continuously bud off the segments. The strobila segments can be divided into immature proglottids (IPs), mature proglottids (MPs) and gravid proglottids (GPs) according to their anteroposterior position and the development of their sexual organs. Just behind the neck are IPs, and further from the posterior are MPs with mature female and male organs and, finally, those whose uterus contains fertilized eggs. The latter proglottids are described as GPs. The gravid or senile terminal proglottids detach or disintegrate [5,6].

RNA-seq technology provides a convenient approach for rapidly characterizing the molecular biological landscape of diverse biological tissues [7]. With the help of RNA-seq, we have gained a preliminary understanding of the mechanism of differentiation of tapeworm segments. On the basis of RNA-seq, Liu et al. [8] reported that homeobox genes and Wnt signalling pathways play core roles in the development and regeneration of *Moniezia expansa*. Li et al. [9] utilized RNA-Seq to determine the transcriptomic profiles of proglottids of *Taenia multiceps* and reported that Hsp90 genes and genes associated with the Wnt, MAPK, and TGF-$\beta$ signalling pathways are important for proglottid differentiation. However, we currently know nothing about the molecular biological mechanisms underlying the differentiation of *S. mansoni* segments. Therefore, there is an urgent need to fill this gap in knowledge, thus laying the foundation for further exploration of the differentiation patterns of tapeworm segments. However, compared with traditional short-read sequencing technologies, nanopore sequencing for full-length transcriptome characterization offers significant advantages: (1) nanopore sequencing can avoid the loss of transcript information from fragmentation, thereby increasing the accuracy of quantification [10]; (2) it enables structural characterization of transcriptional events, facilitating comprehensive molecular mechanism analysis [11]; and (3) it requires minimal sample input, making it particularly advantageous for studies with limited sample availability [12]. In recent years, an increasing number of studies have utilized nanopore sequencing for whole-genome or transcriptome analyses of parasites, such as *Caenorhabditis elegans*, *Clonorchis sinensis*, *Trypanosoma cruzi*, and *Plasmodium falciparum* [13–16].

In this study, we employed nanopore sequencing to perform full-length transcriptome analysis of the SNIPs, MPs, and GPs of *S. mansoni* to comprehensively characterize the differentially expressed genes (DEGs) across different proglottids and screen genes closely related to the segmental differentiation process. Furthermore, previous transcriptomic and proteomic analyses of *S. mansoni* revealed that energy metabolism is significantly increased during the adult stage, suggesting that genes related to energy metabolism may play more important roles in the differentiation and development of adult segments [17,18]. Among the genes related to energy metabolism, pyruvate kinase (PK) is a key regulatory factor that controls metabolic flux and ATP production during glycolysis in parasites. Glycolysis is the core pathway by which cells utilize glucose to produce energy and serves as the primary energy source for parasites [19]. In this process, glucose is converted to pyruvate (PYR) via an oxygen-independent pathway, with the generation of ATP. The multifunctional metabolite PYR participates in subsequent aerobic and anaerobic oxidation, providing the essential energy for cellular activities [20]. The rate of glucose conversion to PYR is coregulated by PK [21]. In the final step of glycolysis, PK irreversibly catalyses the transfer of phosphate from phosphoenolpyruvate (PEP) to adenosine diphosphate (ADP), generating pyruvate and ATP and regulating cellular energy production [22]. To date, PK has been isolated from various parasites, and it plays important roles in parasite–host interactions, parasite survival, and parasite development [23,24].

On the basis of these findings, we aimed to perform comprehensive comparative transcriptomic analyses of different proglottid types to screen genes closely related to energy metabolism in different segments of *S. mansoni* and conduct molecular functional analysis to gain an in-depth understanding of cestode proglottid differentiation. Specifically, the objectives of this study are as follows: (1) identify and characterize DEGs among different proglottids of *S. mansoni* and screen for key molecules involved in the process of segment differentiation, and (2) perform molecular feature and functional analyses of selected key genes to explore their role in tapeworm segment differentiation.

## Materials and methods

### Ethics statement

The animal experiments in this study were carried out based on National Guidelines for Experimental Animal Welfare of China. All animal experimental protocols were approved by the Life Science Ethics Committee of Zhengzhou University (No. ZZUIRB GZR 2022–0062).

### Parasites and animals

The spargana were isolated from infected *Zaocys dhumnades* in Changsha, China [25]. The parasites were identified as *S. mansoni* using the mitochondrial cytochrome c oxidase subunit 1 (*cox*1) gene (S1 Table) via the genotyping method described by Kuchta et al. [1]. Adult worms were obtained as previously described [26]. More specifically, 3 plerocercoids from the same host were perorally administered to a 6-month-old specific pathogen free domestic cat. Forty days after the infection, the cat was preanaesthetized and then euthanized and the adult worms were recovered from the small intestine. The SNIPs, MPs, and GPs were collected according to their positional and morphological characteristics. All the samples were cleaned three times with physiological saline, snap-frozen in liquid nitrogen immediately and stored at –80 °C for subsequent use. Moreover, to obtain related immune serum, 4–6-week-old female Kunming mice (purchased from Henan Experimental Animal Center) were used in this study for recombinant protein immunization. The obtained anti-recombinant protein immune serum was stored at –80 °C for later use. The overall experimental workflow is shown in Fig 1.

### Nanopore sequencing

Total RNA was extracted using TRIzol reagent (Invitrogen, USA). For each proglottid type (SNIP, MP, and GP), three biological replicates were performed, with each replicate sample derived from the same individual tapeworm (totalling three worms). Consequently, three independent RNA samples were obtained for each group. The concentration, purity, and integrity of the extracted RNA were measured with 1% agarose gel electrophoresis, a NanoDrop 2000 (Thermo Fisher Scientific, Wilmington, DE) and an Agilent Bioanalyzer 2100 system (Agilent Technologies, CA, USA). cDNA library construction was initiated with 1 µg of total RNA via a cDNA–PCR sequencing kit (SQK-LSK110 + EXP-PBC096) provided by Oxford Nanopore Technologies (ONT) following the manufacturer's instructions. The final cDNA libraries were added to FLO-MIN109 flow cells and examined on the PromethION platform at Biomarker Technology Company (Beijing, China). The raw data generated from nanopore sequencing were analysed using MinKNOW version 2.2 (Oxford, United Kingdom) with a read quality score ≥ 6 and a read length ≥ 200 bp. Additionally, ribosomal RNA mapped to the rRNA database was also discarded using Minimap2 + SAMtools. The primer sequences on both ends of the clean reads were searched using Pychopper (v.2.4.0) to determine the full-length, nonchimeric (FLNC) sequences. Clusters of FLNC transcripts were obtained after mapping to the reference genome (https://www.ncbi.nlm.nih.gov/datasets/genome/GCA_902702965.1/) with mimimap2, and consensus isoforms were obtained after polishing each cluster with pinfish[2].

### Bioinformatics

Mapped full-length reads with > 5 match quality were chosen for quantification. To ensure that read counts accurately reflect gene expression levels, CPM (counts per million) was employed as the metric for quantifying gene expression [27].

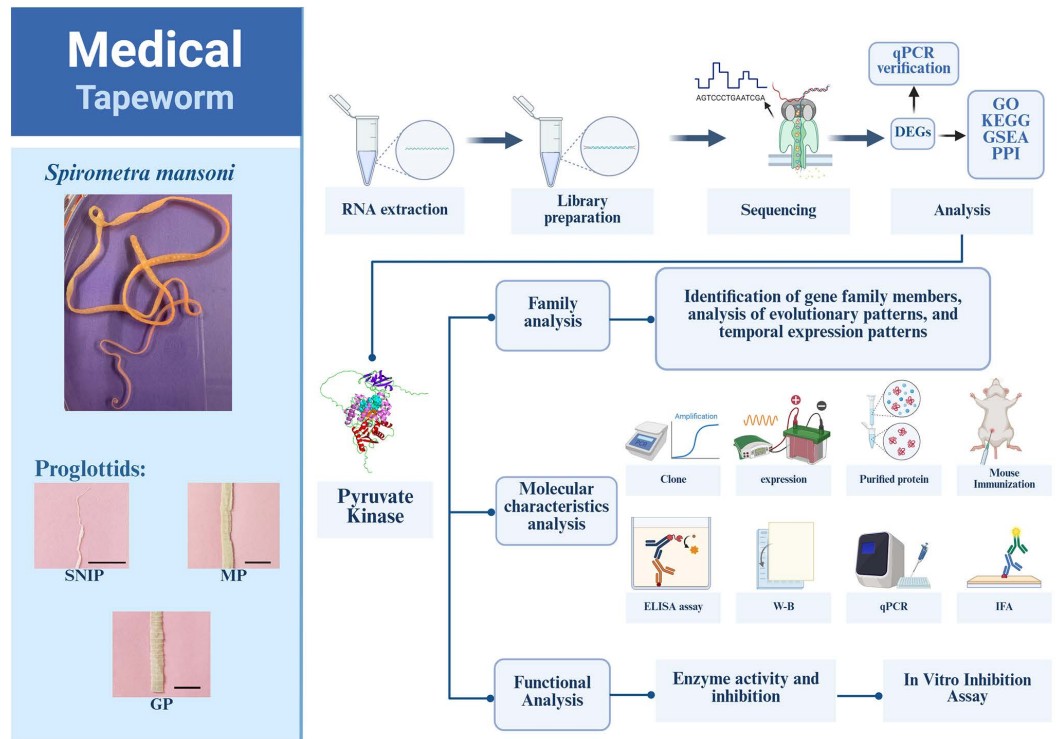

**Fig 1. Flow chart of the study.** First, different proglottids (scolex-neck-immature proglottids, mature proglottids, and gravid proglottids) were prepared for RNA extraction, cDNA library construction, and nanopore sequencing. The differentially expressed genes (DEGs) were subsequently screened and subjected to bioinformatics analysis, and the omics sequencing results were subsequently validated using qPCR. Finally, the pyruvate kinase gene family was selected for in-depth analysis, including analysis of gene family composition, evolutionary patterns, and temporal expression profiles. The molecular characteristics and functional mechanisms of representative family members were further characterized. Flow chart was created in BioRender. candy, **c.** (2025) https://BioRender.com/j04v286.

Differential expression analysis among different proglottids was conducted with the DESeq R package (1.6.3), which is based on a negative binomial distribution model to analyse raw count data. The false discovery rate (FDR) was adjusted and controlled with the method of Benjamini and Hochberg, and DEGs with a log2fold change (FC) ≥ 2 and an FDR < 0.01 were chosen. Databases such as the NR (http://www.ncbi.nlm.nih.gov/), SwissProt (http://www.uniprot.org/), GO (http://www.geneontology.org/), COG (http://www.ncbi.nlm.nih.gov/COG/), KOG (http://www.ncbi.nlm.nih.gov/KOG/), Pfam (http://pfam.xfam.org/), eggNOG (http://eggnogdb.embl.de) and KEGG (http://www.genome.jp/kegg/) databases were used for gene function annotation. Gene ontology enrichment analysis of DEGs was conducted using the GOseq R package-based Wallenius noncentral hypergeometric distribution [28]. KEGG pathway enrichment analysis of DEGs was performed with KEGG Orthology-Based Annotation System (KOBAS) software [29]. Gene set enrichment analysis (GSEA) was conducted using R/clusterProfiler (v4.4.4) to identify gene sets associated with cestode development, with thresholds set as NES > 1, NOM $p$ value < 0.01, and FDR $q$ value < 0.25. Protein–protein interactions (PPIs) for all detected DEGs were predicted using the STRING database (v10.5) with a default threshold of 0.4 and visualized via Cytoscape (v3.10.0) [30].

## Quantitative real-time PCR (qRT–PCR) validation

To confirm the transcription levels of genes identified by RNA-seq, 18 DEGs (3 upregulated and 3 downregulated genes per group) were randomly selected, and their expression levels were measured via qRT–PCR. cDNA was synthesized using a reverse transcription kit (Novoprotein, Shanghai, China) according to the manufacturer's instructions. The primers

used for qRT–PCR were designed according to the RNA-seq data. The gene-specific primers used are listed in S2 Table. The qRT–PCR was conducted with an Applied Biosystems 7500 Fast Real Time PCR System (Applied Biosystems, USA) via the operation protocol described by Liu et al. [4]. The glyceraldehyde-3-phosphate dehydrogenase (GAPDH) gene was used as an endogenous marker to normalize the reactions for the same amplification protocol [26]. After amplification, the relative gene expression levels were analysed according to the comparative $2^{-\Delta\Delta CT}$ method [31]. For all the quantitative PCR analyses for each gene, three biological replicates were used, and differences were considered statistically significant at $p < 0.05$.

### Identification of *Sm*PK family members

PK plays a key role in the production of ATP during glycolysis. Glycolysis is crucial for cellular energy metabolism and serves as the primary energy source for parasites [19]. Furthermore, inhibiting PK activity or gene expression can interfere with the energy metabolism of parasites, thereby affecting their growth and development [32]. Owing to the limited research on this enzyme in cestodes, it is unclear whether it plays an important role in the development of tapeworm segments. Furthermore, comparative transcriptomic analysis revealed that PK plays key roles in glycolysis. Therefore, we selected the PK gene family for further analysis to explore its effects on the differentiation and development of tapeworm segments.

Candidate sequences containing the PK protein domain were searched using the NCBI Conserved Domains database (www.ncbi.nlm.nih.gov/Structure/cdd/wrpsb.cgi). All candidate *Sm*PK sequences were obtained from the WormBase ParaSite database (https://parasite.wormbase.org) and recently published transcriptomic and proteomic data. These extracted sequences were identified as members of the PK family through querying for genes annotated with the Pfam domain accessions pfam00224 and pfam02887. All screened candidates from the multiomic data were analysed with the HMMER tool to confirm the presence of the conserved PK [33]. The basic physical and chemical properties of all the identified *Sm*PKs were determined according to previously described methods [34]. To precisely map *Sm*PK genes on scaffolds, distribution information was collected from the *S. erinaceieuropaei* draft genome (NCBI accession GCA_902702965.1), and gene distribution maps were generated using TBtools2 (v2.136) software. Gene duplication events within the *Sm*PK gene family were identified using TBtools2, along with whole-genome and GFF files. Comparative synteny analysis of PK genes between *S. mansoni* and other cestodes was also performed using TBtools2.

### qRT–PCR analysis

Quantitative RT–PCR (qRT–PCR) analysis was performed to monitor the expression levels of the identified *Sm*PKs in three stages of the *S. mansoni* life cycle: the plerocercoid stage, the adult stage (including the SNIPs, MPs and GPs), and the egg stage. The gene-specific primers used are listed in S3 Table. The qRT–PCR procedure was performed as described above. Three biological replicates were used, and differences were considered statistically significant at $p < 0.05$.

### Phylogenetic analysis

All available PK sequences of medically important cestodes and trematodes were also extracted from the WormBase ParaSite database. The multiple expectation maximization for motif elicitation (MEME) method was used to analyse potential motifs [35]. Using MAFFT v7, multiple protein sequence alignment was carried out [36]. The Bayesian inference (BI) and maximum likelihood (ML) methods were used to conduct phylogenetic analysis. BI analysis was performed in MrBayes v.3.2 [37]. The analysis consisted of two independent runs, each with four Markov chain Monte Carlo (MCMC) chains running for 5 000 000 generations and sampling every 100th generation. ML analysis was performed in MEGA v.7.0 [38]. The confidence in each node was assessed by bootstrapping (1000 pseudoreplicates).

## Cloning, expression, and characterization of r*Sm*PK1

PPI analysis revealed that *Sm*PK1 occupies a central node and may play a more significant role in glycolysis. Therefore, *Sm*PK1 was selected as a representative family member for cloning and characterization. The target gene was amplified with specific primers carrying BamHI and PstI sites (forward, 5′-TAGGATCCATGACTGGGGCTT TGAAAGTTCTC-3′; reverse, 5′-TACTGCAGTCACTTGCAGACCTGAATCG-3′). The cycling protocol was as follows: 30 cycles of 95 °C for 50 s, 58 °C for 50 s and 72 °C for 50 s. The PCR products were purified, digested, cloned and inserted into the pQE-80L vector (Ipswich, USA). The recombinant plasmid was then transformed into *Escherichia coli* BL21 (Novagen, La Jolla, CA, USA). The expression of r*Sm*PK1 was induced by the addition of 0.2 mM IPTG, and the cells were then cultured at 25 °C for 13 h. r*Sm*PK1 was purified by Ni²⁺ affinity chromatography (Shenggong Biotech, Shanghai, China) and identified via SDS–PAGE. Images of the gels were recorded via an ImageScanner (GE Healthcare, Fairfield, CT). Another gel was prepared by the same method and used for Western blot analysis. The antibody titres in the sera of immunized mice were measured using indirect ELISA, and the localization of the target gene was determined by IFA.

## Enzyme activity assay

The enzyme activity of r*Sm*PK1 was assayed via the 2,4-dinitrophenylhydrazine method. The reaction mixture composition and incubation conditions were adapted from Yue et al. [24]. The optimum *in vitro* catalytic conditions for r*Sm*PK1 were determined by changing the concentration of r*Sm*PK1, the reaction temperature and the pH of the buffer solution. The effects of metal ions (0.3 mM $Zn^{2+}$, $Ca^{2+}$, $Cu^{2+}$, or $Mn^{2+}$) and tannic acid on PK activity were assessed. Michaelis–Menten kinetics were determined by ADP/PEP titration under optimized conditions.

## *In vitro* inhibition assay

Two adult cestodes with similar individual characteristics were sequentially washed three times each with PBS and RPMI 1640 medium to remove contaminants. The adult cestodes were then divided into an experimental group (100 µM tannic acid treatment) and a control group (inhibitor-free incubation). After incubation at 37 °C, 4 uniformly sized GPs were collected from the posterior ends of both the experimental and control cestodes at 3-h intervals (3/6/9 h) and placed into prelabelled centrifuge tubes. Finally, according to the kit instructions, the tissue samples were weighed and processed for subsequent quantification of pyruvate (Sangon Biotech, Shanghai, China), total sugars (Sangon Biotech, Shanghai, China), and triglycerides (Jiancheng Biotech, Nanjing, China). Additionally, the enzyme activity of *Sm*PK1 in GPs was detected using the 2,4-dinitrophenylhydrazine chromogenic method.

# Results

## Transcriptomic analysis

To evaluate the reliability of the experimental data and patterns of intergroup differences, principal component analysis (PCA) was performed on the sequencing data in this study. The results (S1 Fig) revealed that biological replicates within the same group clustered tightly together, while distinct separation was observed between different groups, with no abnormal outliers detected. The cDNA libraries of various proglottids of *S. mansoni*, namely, SNIPs, MPs, and GPs, were sequenced via Oxford Nanopore Technology. An average of 7.55 G bp of clean data for nine cDNA libraries was obtained. The average N50 length was 1,167 bp, with a mean length of 1,027 bp. After rRNA was filtered out, an average of 5,799,182 clean reads and an average of 5,421,673 full-length reads were generated. The average full-length sequence ratio was 93.48%, and the average mapping rate was 97.70% (Table 1). Finally, 26,039 redundancy-filtered transcript sequences were obtained with an average N50 length of 1,583 bp, a mean length of 1,194 bp, and an average maximum length of 7,655 bp through the merging of the consensus sequences (S4 Table).

**Table 1. Statistics of full-length sequence data.**

| Sample | Number of clean reads (except rRNA) | Number of full-length | Full-length percentage | Number of Mapped reads | Mapped rate |
|---|---|---|---|---|---|
| SNIP1 | 5,722,478 | 5,335,328 | 93.23% | 5,234,151 | 98.10% |
| SNIP2 | 6,168,349 | 5,757,834 | 93.34% | 5,650,575 | 98.14% |
| SNIP3 | 4,682,765 | 4,372,624 | 93.38% | 4,292,112 | 98.16% |
| MP1 | 5,889,671 | 5,483,313 | 93.10% | 5,371,695 | 97.96% |
| MP2 | 6,948,179 | 6,528,473 | 93.96% | 6,392,875 | 97.92% |
| MP3 | 5,274,214 | 4,938,049 | 93.63% | 4,842,047 | 98.06% |
| GP1 | 5,571,157 | 5,218,680 | 93.67% | 5,058,880 | 96.94% |
| GP2 | 6,143,529 | 5,732,686 | 93.31% | 5,560,457 | 97.00% |
| GP3 | 5,792,299 | 5,428,071 | 93.71% | 5,266,339 | 97.02% |
| Mean | 5,799,182 | 5,421,673 | 93.48% | 5,296,570 | 97.70% |

## Screening of DEGs

A total of 16,501 genes were identified across the three different proglottids (SNIPs, MPs, and GPs), including 1,431 novel genes and 4,486 DEGs. Among the 4,486 DEGs, 1,498 were differentially expressed between SNIPs and MPs (S5 Table), 3,267 were differentially expressed between MPs and GPs (S6 Table), and 3,686 were differentially expressed between SNIPs and GPs (S7 Table). Additionally, 673 DEGs were coexpressed across all three proglottids (S8 Table and S2a Fig). Heatmap analysis revealed that the overall expression of DEGs between MPs and GPs was highly similar, whereas significant differences in DEG expression were observed in comparison with those in SNIPs (S2b Fig). The volcano plots revealed that 1,290 genes were upregulated and that 208 genes were downregulated between SNIPs and MPs (S2c Fig). Among the MP and GP genes, 1,986 genes were upregulated, whereas 1,281 genes were downregulated (S2d Fig). Among the SNIPs and GPs, 2,544 genes were upregulated, and 1,142 genes were downregulated (S2e Fig).

## GO analysis of DEGs

GO analysis was conducted on the 673 coexpressed DEGs, resulting in the successful annotation of 267 DEGs into 47 subcategories. Among these DEGs, the most annotated subcategory in the biological process (BP) category was cellular process (52.8%). In the molecular function (MF) category, the predominant subcategories included binding (49.43%) and catalytic activity (37.07%). In the cellular component (CC) category, the most annotated subcategories were cell (31.08%) and cell part (31.18%) (Fig 2A). Further enrichment analysis revealed that, within the BP category, DEGs were enriched primarily in cilium organization, cilium movement, and glycolytic process. In the CC category, significant enrichment was observed for the cilium and motile cilium, whereas in the MF category, the main enriched terms were metal ion binding and GTP binding (Fig 2B). Among the different proglottids, 596 DEGs between SNIPs and MPs were successfully annotated to 49 subcategories (S3 Fig), 1486 DEGs between MPs and GPs were successfully annotated to 54 subcategories (S4 Fig), and 1655 DEGs between SNIPs and GPs were successfully annotated to 55 subcategories (S5 Fig).

## KEGG analysis of DEGs

KEGG analysis of the 673 DEGs revealed that 101 DEGs were annotated to 84 pathways, with the largest number of DEGs associated with the glycolysis pathway (Fig 2C). Further enrichment analysis revealed that the glycolysis pathway was the most significantly enriched pathway (Fig 2D). Among the different proglottids, 193 DEGs between SNIPs and MPs were annotated to 107 pathways, 604 DEGs between MPs and GPs were annotated to 168 pathways, and 642 DEGs between SNIPs and GPs were annotated to 173 pathways (S6–S8 Figs). Given that the glycolysis pathway exhibited

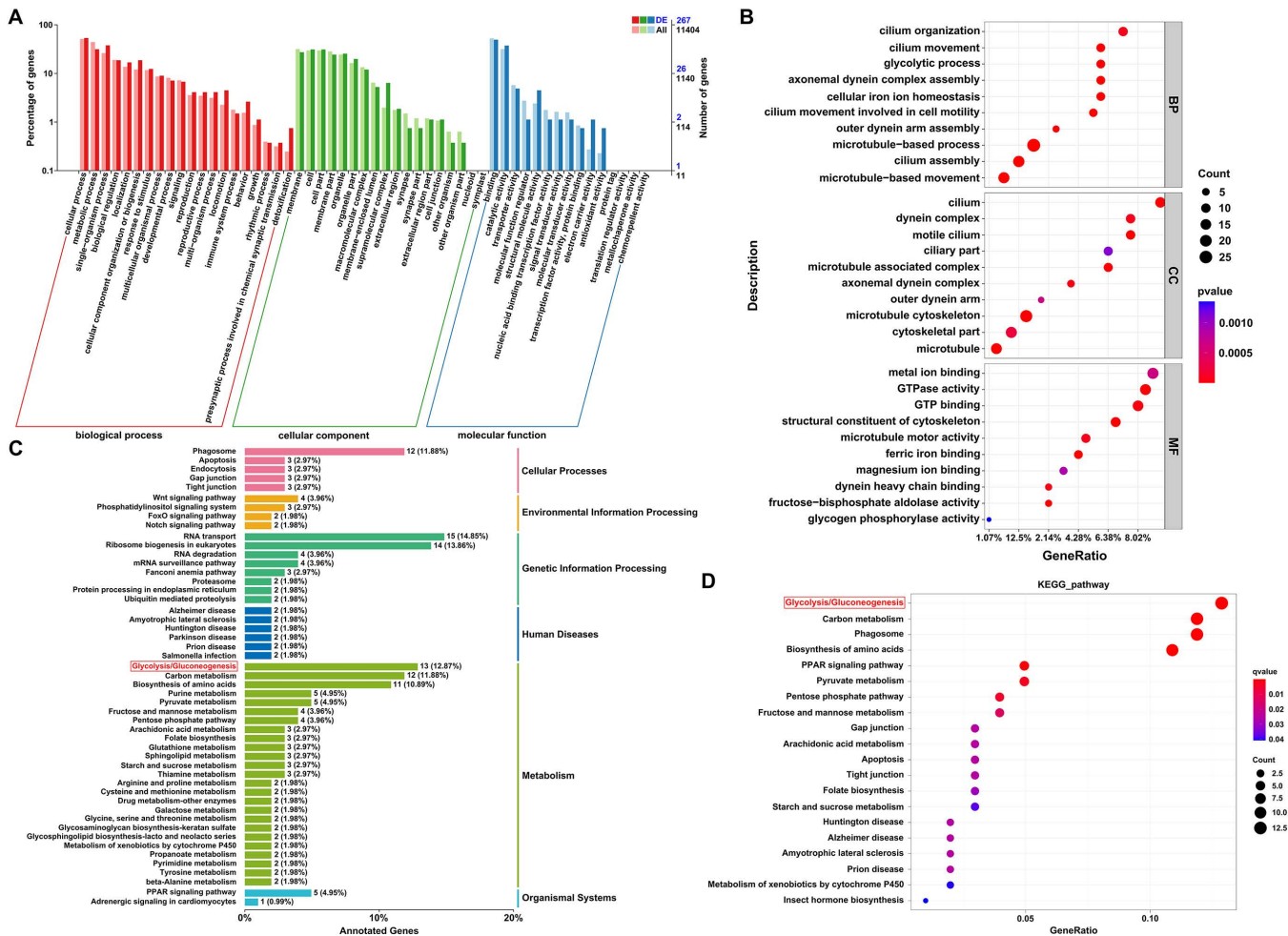

**Fig 2. GO and KEGG analysis of coexpressed differentially expressed genes among different segments of *S. mansoni*. A.** GO subcategories of coexpressed DEGs across three proglottids. **B.** Top 10 enriched terms of coexpressed DEGs in three proglottids across the BP, CC, and MF categories. **C.** KEGG classification diagram of coexpressed DEGs across three proglottids. **D.** Top 20 significantly enriched KEGG pathways of coexpressed DEGs across three proglottids.

the greatest enrichment, it was selected as a representative pathway for further investigation. Genes associated with the glycolysis pathway were first screened in SNIPs vs. MPs and MPs vs. GPs. These genes were subsequently ranked on the basis of their expression levels in different proglottids, and the GSEA method was employed to systematically evaluate the activity of this pathway across the proglottids, revealing the dynamic regulatory characteristics of glycolysis during proglottid development. Among the SNIPs and MPs, seven key differentially expressed proteins (DEPs) were significantly upregulated the MPs (Fig 3A). The GSEA results revealed significant positive enrichment of glycolysis pathway genes in MPs (NES = 1.345, $p < 0.01$), suggesting increased pathway activity (Fig 3B). Heatmap analysis revealed that the DEGs in the glycolysis pathway were generally upregulated in the MPs, which is consistent with the GSEA results, indicating that this pathway plays a central role in MPs (Fig 3C). The glycolysis pathway was enriched in both MPs and GPs, with greater enrichment observed in GPs (S9 Fig). In summary, KEGG analysis revealed significant enrichment of the glycolysis pathway, and GSEA further indicated that this pathway was more active in the MPs and GPs.

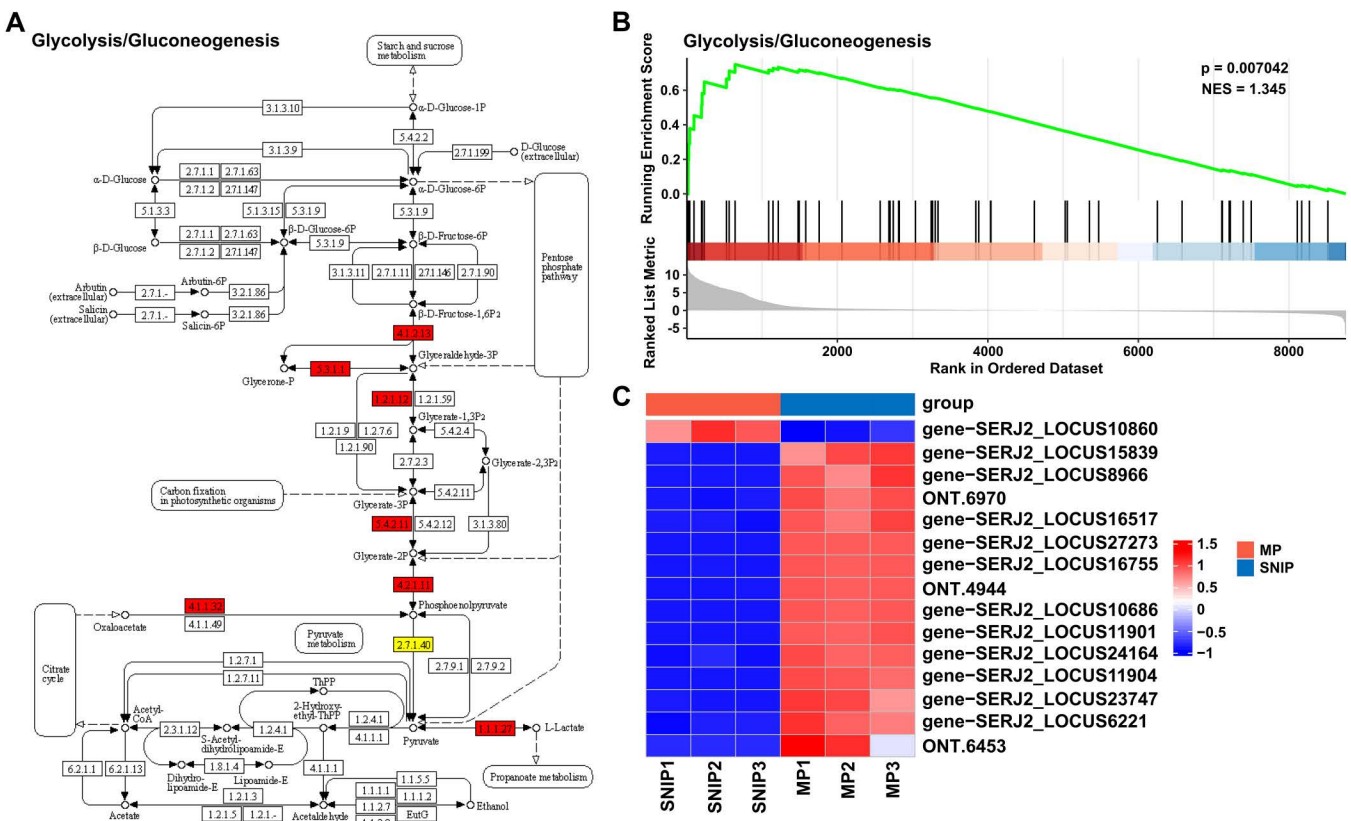

**Fig 3. Analysis of the glycolysis pathway in SNIPs and MPs. A.** Fifteen DEGs involved in the glycolytic pathway were identified in the SNIPs and MPs. The numbers in boxes represent genes. Red symbols represent genes upregulated in MPs. Yellow symbols represent genes with mixed regulation. **B.** Gene set enrichment analysis of the glycolysis pathway in the SNIPs and MPs. A positive ES indicates significant enrichment of this pathway in MPs. **C.** Heatmap of 15 DEGs in the glycolysis pathway.

## PPI analysis of DEGs

A total of 39 metabolism-related genes were screened and submitted to the STRING database for PPI network analysis (S9 Table). PPI analysis revealed three key hub genes encoding PK (SERJ2_LOCUS24164, 12 nodes), glyceraldehyde-3-phosphate dehydrogenase (SERJ2_LOCUS10686, 12 nodes), and glutamate dehydrogenase (ONT.2601, 12 nodes) (Fig 4A). These genes play critical roles in biological processes such as glycolysis, carbon metabolism, amino acid biosynthesis, and metabolism. A total of 67 metabolism-related genes were identified between SNIPs and MPs, with key hub genes encoding PK (18 nodes), enolase (SERJ2_LOCUS15839, 18 nodes), and malate dehydrogenase (SERJ2_LOCUS9530_568, 18 nodes). These genes were upregulated in MPs (S10 Table and Fig 4B). Among the 202 screened metabolism-related genes in MPs and GPs, PK was located at the central node and was upregulated in GPs (S11 Table and Fig 4C). Among SNIPs and GPs, 226 metabolism-related genes were screened out, and the key hub genes included PK, which was also upregulated in GPs (S12 Table and Fig 4D). Therefore, in both the coexpression group and the different segments, PK was consistently in the core hub position, and its expression level increased with the development and maturation of the segments, suggesting that PK plays a more prominent role in segment differentiation.

PLOS Neglected Tropical Diseases

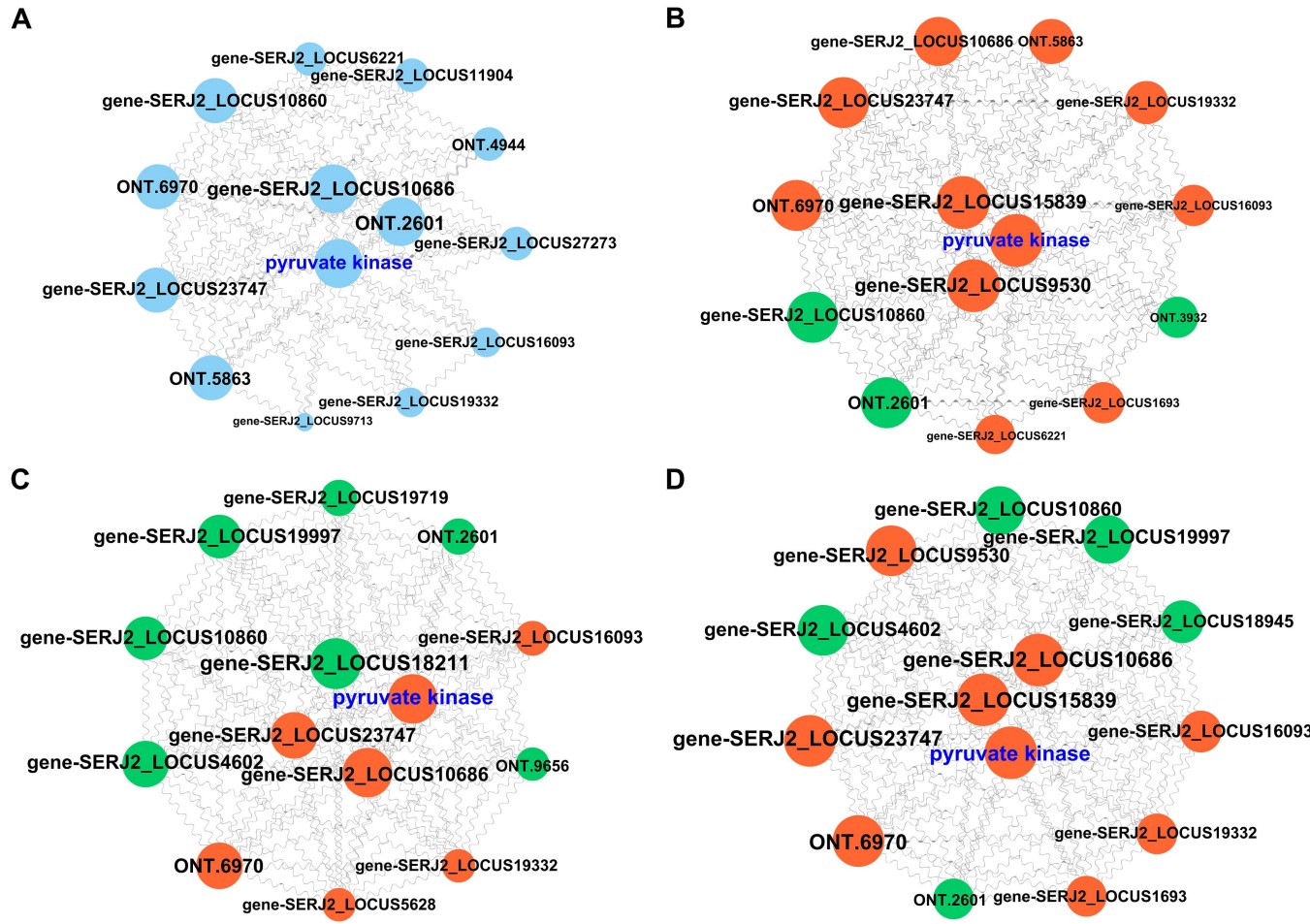

**Fig 4. Protein–protein interaction network analysis of metabolism-related DEGs. A.** PPI network of coexpressed DEGs across three proglottids. The blue nodes represent DEGs. **B.** PPI network of DEGs between SNIPs and MPs. **C.** PPI network of DEGs between MPs and GPs. **D.** PPI network of DEGs between SNIPs and GPs. Red and green nodes represent upregulated and downregulated genes, respectively.

## qRT–PCR validation

A significant positive correlation was observed between the nanopore sequencing results and the results of the qPCR-based quantification of gene expression levels ($R^2 = 0.6471$), confirming the high reliability of the RNA-seq data (S10 Fig). Specifically, between SNIPs and MPs, the expression of PKLR, Lipm, and DUF5734 was upregulated in SNIPs (Fig 5A), whereas that of HSP70, PKM, and ENO was upregulated in MPs (Fig 5B). Among the MPs and GPs, Rars, AKR1E2, and Glud were upregulated in MPs (Fig 5C), whereas K02A2.6, FABP2, and DCT were upregulated in GPs (Fig 5D). Among SNIPs and GPs, the putative ACBP, Glud, and PKLR genes were upregulated in SNIPs (Fig 5E), whereas FABP2, PKM, and ENO were upregulated in GPs (Fig 5F). The qPCR results revealed expression trends consistent with the RNA-seq data, further validating the reliability of the transcriptomic analysis results.

## Screening of the *Sm*PK gene family

Through transcriptomic analysis, we found that the glycolysis pathway plays a critical role in the differentiation of *S. mansoni* proglottids. Within this pathway, PK occupies a central node and has high expression levels. Additionally, previous

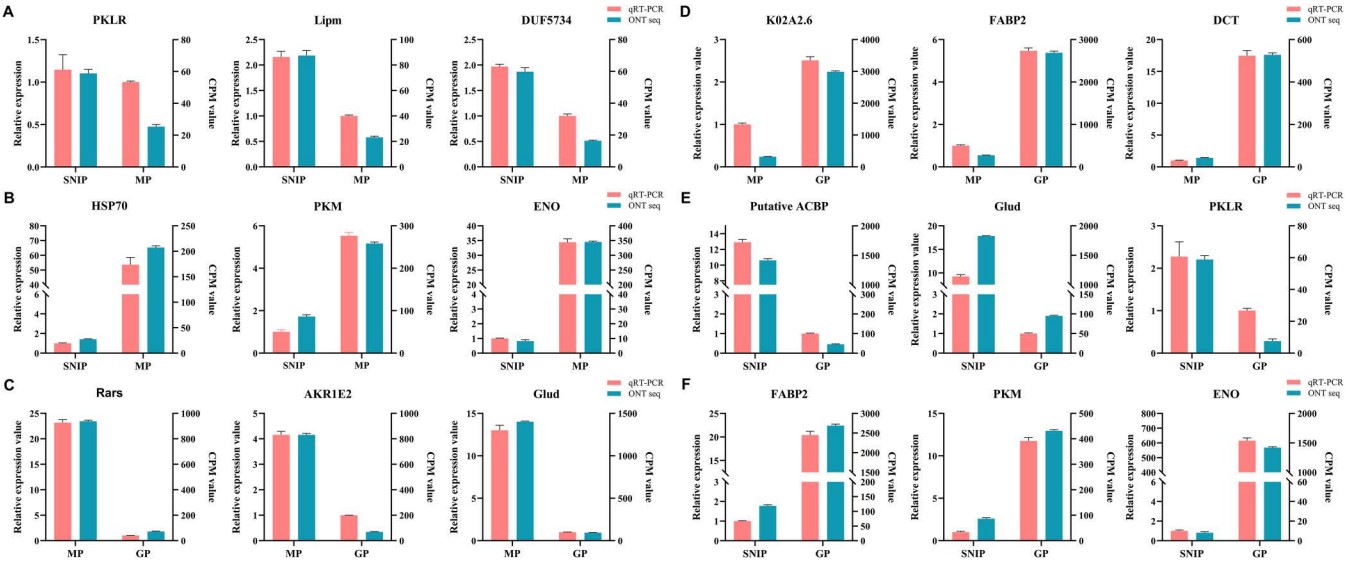

**Fig 5. qPCR validation of DEGs. A** and **B** represent the upregulated genes in SNIPs and MPs, respectively. **C** and **D** represent the upregulated genes in MPs and GPs, respectively. **E** and **F** represent upregulated genes in SNIPs and GPs, respectively. GAPDH was used for normalization. The results are presented as the means±SEMs (standard means of error) of the samples (n = 3).

studies have demonstrated the essential role of PK in the survival and development of parasites. Therefore, the *Sm*PK gene family was selected for further analysis.

On the basis of the transcriptome data, four PK family members (*Sm*PK1–*Sm*PK4) were identified in *S. mansoni*. The length of the *Sm*PKs ranged from 4,725 bp to 15,810 bp. The predicted number of amino acids ranged from 228 aa to 603 aa. The domain lengths ranged from 118 aa to 351 aa (Table 2). Scaffold-level localization revealed that *Sm*PK1 is located on scaf3, *Sm*PK2 is located on scaf2, *Sm*PK3 is located on scaf4, and *Sm*PK4 is located on scaf1 (Fig 6A). Conserved domain analysis revealed that all the *Sm*PKs contained the PK conserved domain (Fig 6B). Exon–intron structure analysis revealed that *Sm*PK1 contains 10 exons, *Sm*PK2 contains 9 exons, *Sm*PK3 contains 5 exons, and *Sm*PK4 contains 8 exons (Fig 6C). To explore the expansion and contraction events of *Sm*PKs during evolution, gene duplication analysis was conducted, which revealed no duplication events among the *Sm*PK genes (Fig 6D). Synteny analysis revealed a lack of collinearity between *S. mansoni* and both Taeniidae cestodes and the more closely related pseudophyllidean cestodes (Fig 6E and 6F). To determine the expression patterns of the identified *Sm*PKs, we sampled eggs, plerocercoids, and different adult proglottids for analysis via qRT–PCR (Fig 6G). In the egg, plerocercoid, and adult stages, three *Sm*PKs were expressed. In the adult stage, two genes were highly expressed. Notably, these two highly expressed genes were markedly upregulated in both MPs and GPs.

**Table 2. Annotation characteristics of *Sm*PKs.**

| Genes | Gene ID | Protein length (aa) | PK domain coordinates | Domain length (aa) |
|-------|---------|---------------------|----------------------|--------------------|
| *Sm*PK1 | SERJ2_LOCUS24164 | 603 | 99-449, 465-582 | 351, 118 |
| *Sm*PK2 | SERJ2_LOCUS10860 | 597 | 49-397, 417-534 | 349, 118 |
| *Sm*PK3 | SERJ2_LOCUS27030 | 228 | 1-124 | 124 |
| *Sm*PK4 | SERJ2_LOCUS10243 | 443 | 18-325 | 308 |

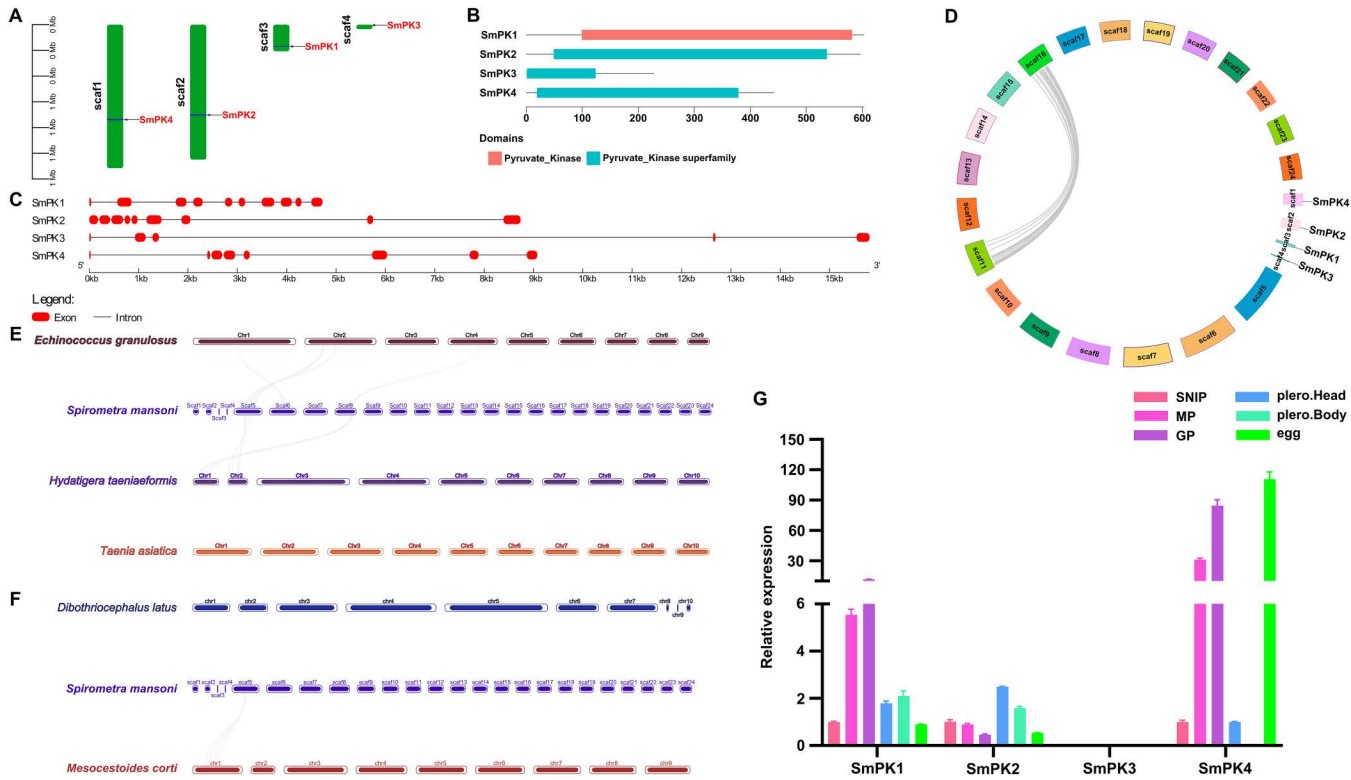

**Fig 6. Analysis of the gene structure and expression patterns of *Sm*PKs. A.** Scaffold-level localization: the green columns represent different scaffolds of *S. erinaceieuropaei* (GCA_902702965.1). **B.** Analysis of conserved domains: Red and blue rectangles represent conserved domains of pyruvate kinase and its subfamily, respectively. **C.** Intron–exon structure: the red rounded rectangles represent exons; the black lines connecting two exons represent introns. **D.** Collinearity analysis of *Sm*PKs: the grey shading indicates the gene density. **E** and **F** Collinearity analysis of the PKs among medically important cestodes. The grey shading indicates syntenic regions of *S. mansoni* with other medically important cestodes in non-PK gene families. **(E)** Collinearity analysis between *S. mansoni* and *Echinococcus granulosus*, *Hydatigera taeniaeformis*, and *Taenia asiatica*. **(F)** Collinearity analysis between *S. mansoni* and *Dibothriocephalus latus* and *Mesocestoides corti*. **G.** PK gene expression of *S. mansoni* at different stages determined via qRT–PCR. SNIP: scolex-neck-immature proglottid; MP: mature proglottid; GP: gravid proglottid; Plero.Head: plerocercoid head; Plero.Body: plerocercoid body. GAPDH was used as an internal reference gene. The expression level was measured with the $2^{-\Delta\Delta CT}$ method. The data were averaged from three repeats, and the error bars represent the SDs ($n = 3$).

## Phylogenetic patterns of PKs in zoonotic cestodes and trematodes

A total of 172 PK sequences were retrieved from 16 medically important cestodes and 25 medically important trematodes (S13 Table). All the PK sequences from the cestodes and trematodes were divided into two major clades: Clade I and Clade II (Fig 7). Specifically, Clade I consisted of a cestode group (Group A) and a trematode group (Group B). In Group A, partial sequences from the families Taeniidae and Hymenolepididae formed monophyletic branches. In Group B, sequences from the families Opisthorchiidae, Paragonimidae, Fasciolidae, and Schistosomatidae also formed monophyletic branches. Clade II similarly comprised a cestode group (Group C) and a trematode group (Group D). In Group C, sequences from the families Taeniidae, Diphyllobothriidea, and Hymenolepididae formed monophyletic branches, whereas in Group D, sequences from the families Opisthorchiidae, Paragonimidae, and Fasciolidae clustered together. Interestingly, although the PKs of zoonotic cestodes and trematodes presented high sequence diversity, the PKs from the same family tended to cluster together, indicating a certain degree of conservation of PKs. For the PK sequences in *S. mansoni*, *Sm*PK1 and *Sm*PK3 were distributed in Group A of Clade I, and *Sm*PK2 and *Sm*PK4 were inserted into Group C of Clade II.

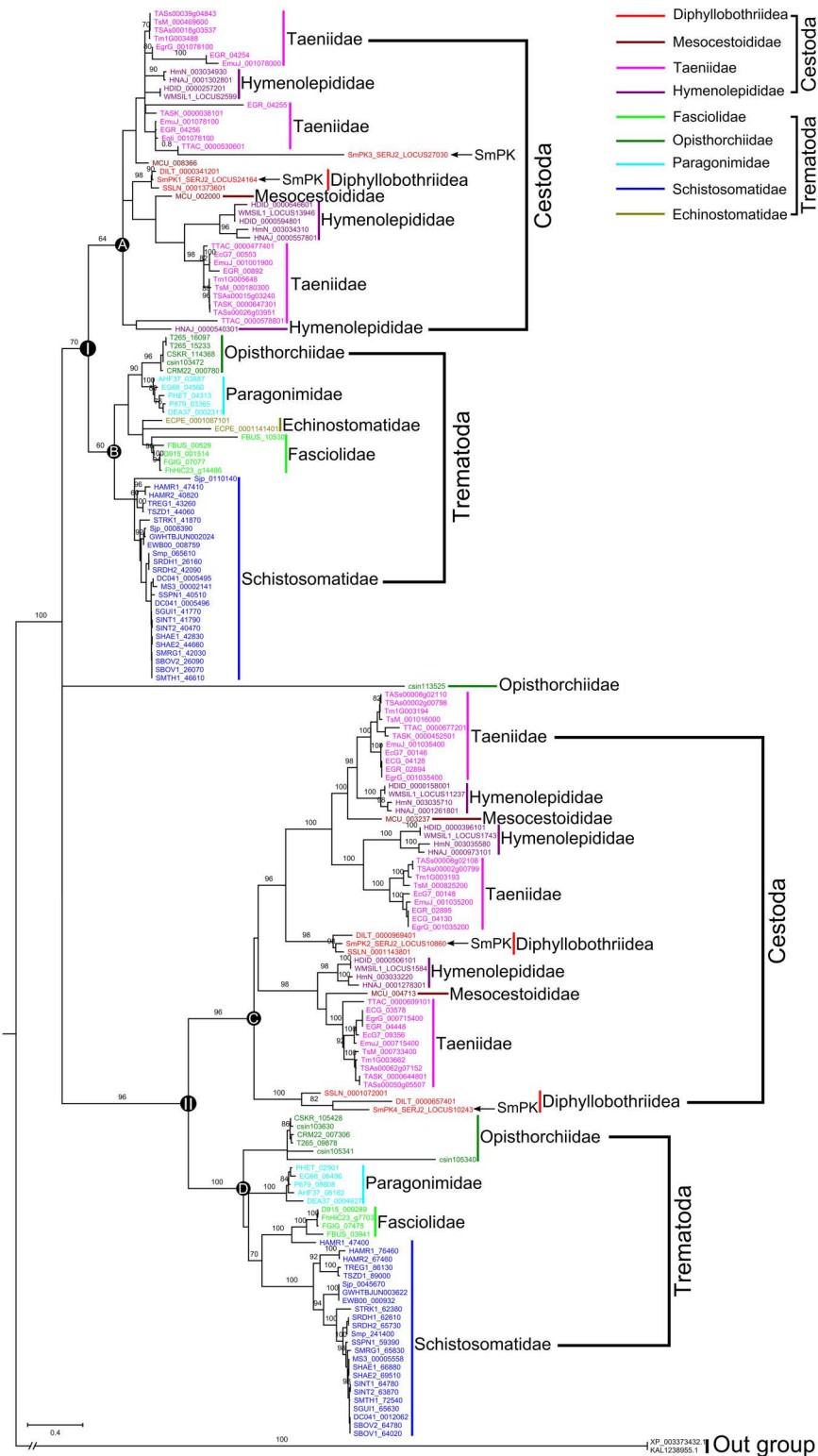

**Fig 7. Phylogenetic analysis of pyruvate kinases in medically important cestodes and trematodes via the maximum likelihood method.** Nematodes were selected as the outgroup, and the values on the branches represent the bootstrap values.

## Molecular characterization of *Sm*PK1

Given its central role in the PPI network and high expression level, *Sm*PK1 was selected as a representative family member for molecular characterization. *Sm*PK1 is a cytoplasmic protein with a Mw of 65,237.82 Ku and an isoelectric point of 6.96 (S11 Fig). mRNA transcription (1812 bp) of the *Sm*PK1 gene was detected in eggs, plerocercoids (both head and body regions) and different proglottids (SNIPs, MPs, GPs) in adult worms. qPCR analysis revealed that the transcription level of *Sm*PK1 in GPs was the highest, followed by MPs and the posterior region of plerocercoids (Fig 8A and 8B). The optimal induction conditions for r*Sm*PK1 were 25 °C with 0.2 mM IPTG for 13 h (Fig 8C–8E). r*Sm*PK1 was purified using Ni²⁺ affinity chromatography and identified via SDS–PAGE (Fig 8F). The optimal protein concentration was 0.0625 µg/mL, and 1: 100 was the best mouse serum dilution (Fig 8G). The cut-off value of the standard for subsequent tests was 0.1234

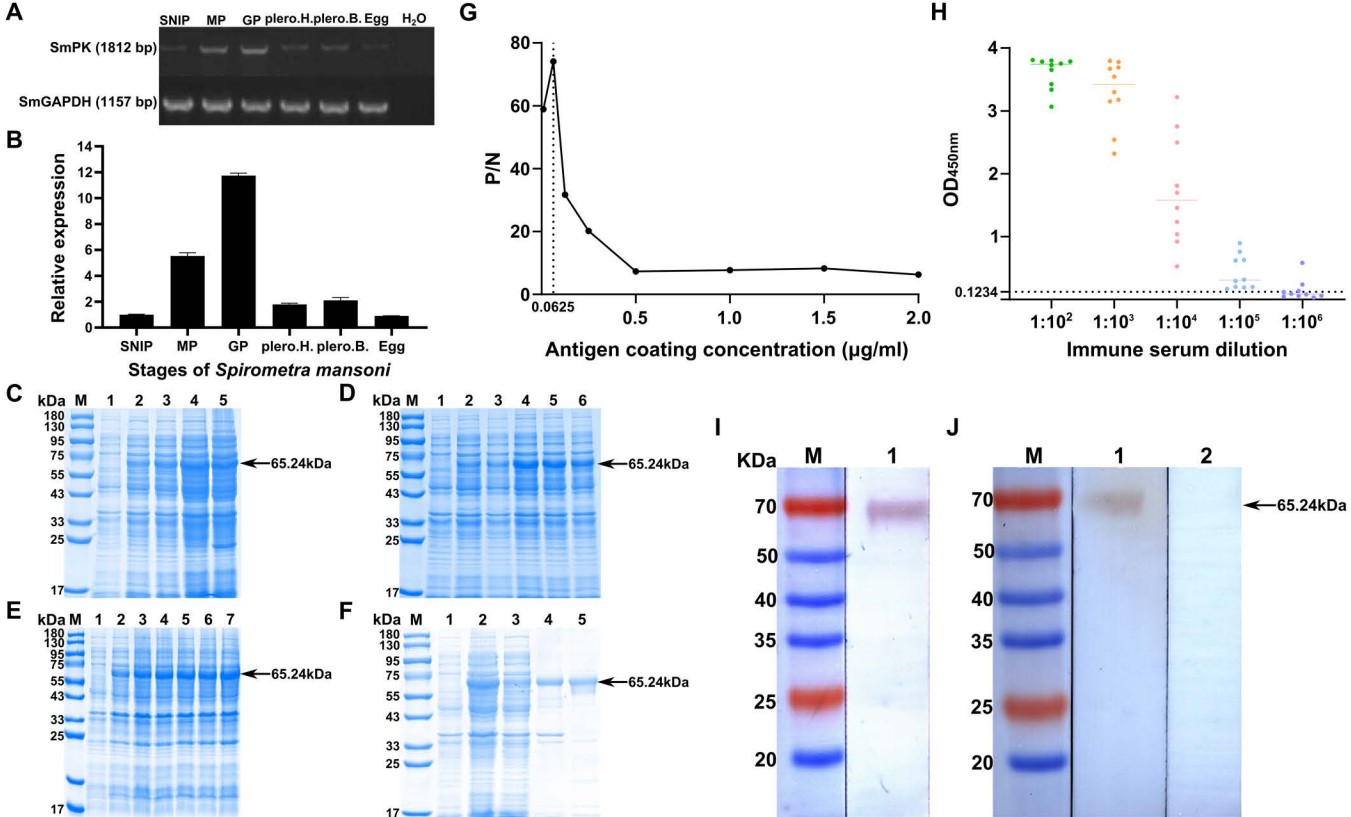

**Fig 8. Molecular characterization of *Sm*PK1. A, B** Transcription patterns of the PK gene in various developmental stages of *S. mansoni*, including eggs, plerocercoids, and adults. **(A)** Conventional RT–PCR. **(B)** Real-time RT–PCR. The housekeeping gene (GAPDH) was used as an internal reference. H₂O was used as a negative control. plero.H.: plerocercoid head; plero.B.: plerocercoid body; SNIP: scolex-neck-immature proglottid; MP: mature proglottid; GP: gravid proglottid; Egg: eggs in the gravid proglottid uterus. **C.** Temperature gradient-induced recombinant protein expression. M: prestained protein marker; 1: uninduced control; 2: induced at 16 °C; 3: induced at 20 °C; 4: induced at 25 °C; 5: induced at 30 °C. **D.** IPTG concentration gradient-induced recombinant protein expression. M: prestained protein marker; 1: 0 mM IPTG; 2: 0.05 mM IPTG; 3: 0.1 mM IPTG; 4: 0.2 mM IPTG; 5: 0.4 mM IPTG; 6: 0.5 mM IPTG. **E.** Time course induction of recombinant protein expression. M: prestained protein marker; 1: uninduced control; 2: 11 h induction; 3: 12 h induction; 4: 13 h induction; 5: 14 h induction; 6: 15 h induction; 7: 16 h induction. **F.** Solubility analysis and purification of r*Sm*PK1. M: Prestained protein marker; 1: uninduced bacterial cultures; 2: the lysate of the induced recombinant bacteria harbouring pQE-80 L-r*Sm*PK1 after ultrasonication; 3: the protein in the supernatant; 4: the protein in the precipitate; 5: r*Sm*PK1 purified by the Ni-NTA-Sefinose column. **G.** Determination of the optimal antigen coating concentration. **H.** Anti-r*Sm*PK immunoserum potency assay. Green, orange, pink, blue, and purple represent serum dilutions of 1:10², 1:10³, 1:10⁴, 1:10⁵, and 1:10⁶, respectively. r*Sm*PK1 antigenicity analysis. **I.** M: prestained protein marker; 1: r*Sm*PK1 + anti-r*Sm*PK1 serum. **J.** M: prestained protein marker; 1: r*Sm*PK1 + infected mouse serum; 2: r*Sm*PK1 + normal mouse serum.

(Fig 8H). Western blotting analysis confirmed that rSmPK1 was recognized by both infected serum and anti-rSmPK1 serum (Fig 8I and 8J). The immunolocalization test revealed specific fluorescence staining in eggshells, plerocercoids and adults. SmPK1 was concentrated in the cortex and parenchyma of plerocercoids and adults, and a prominent fluorescence signal was observed in the tissues and follicles around the uterus in mature or GPs and eggshells, which indicates that PK was highly expressed in the uterus and eggshell (Fig 9).

## Enzymatic reaction kinetics of rSmPK1

The enzymatic activity of rSmPK1 gradually increased with increasing rSmPK1 concentration and stabilized at a concentration of 20 ng/μL (Fig 10A). The optimum temperature of rSmPK1 for catalysing the substrate reaction was 37 °C, and the optimum buffer pH was 8.0 (Fig 10B and 10C). The metal ions $K^+$ and $Mg^{2+}$ significantly enhanced the enzyme activity of rSmPK1, whereas $Mn^{2+}$ inhibited the enzyme activity of rSmPK1 (Fig 10D). The suppressive effect of tannic acid ($r = 0.953$, $p < 0.0001$) on rSmPK1 activity was dose dependent, with complete inhibition achieved at 8 μM tannic acid (Fig 10E). The enzymatic reaction conformed to simple Michaelis–Menten kinetics (Fig 10F). The kinetic parameter Vmax of PEP was 1.303 mM/min, and the km value was 1.272 mM (Fig 10G). The kinetic parameter Vmax of ADP was 0.95 mM/min, and the km value was 0.54 mM (Fig 10H).

## In vitro inhibition assay

To elucidate the impact of SmPK1 on the energy metabolism of S. mansoni, we performed an in vitro inhibition assay using tannic acid inhibitor (Fig 10I and 10J). Quantitative analysis revealed significant dynamic changes in the metabolic activity of adult proglottids after treatment with 100 μM tannic acid for 3–9 h, as follows: 1. The activity of PK was continuously inhibited, with a reduction of 31.87–39.78% ($P < 0.001$), indicating that tannic acid effectively and sustainably inhibits SmPK1 activity (Fig 10K). 2. The levels of pyruvate, the catalytic product of PK, decreased synchronously by 31.95–39.17% ($p < 0.001$), confirming that the inhibition of PK activity obstructs the glycolytic pathway and further validating the critical role of SmPK1 in glycolysis (Fig 10L). 3. Triglycerides were consumed in a time-dependent manner, with a

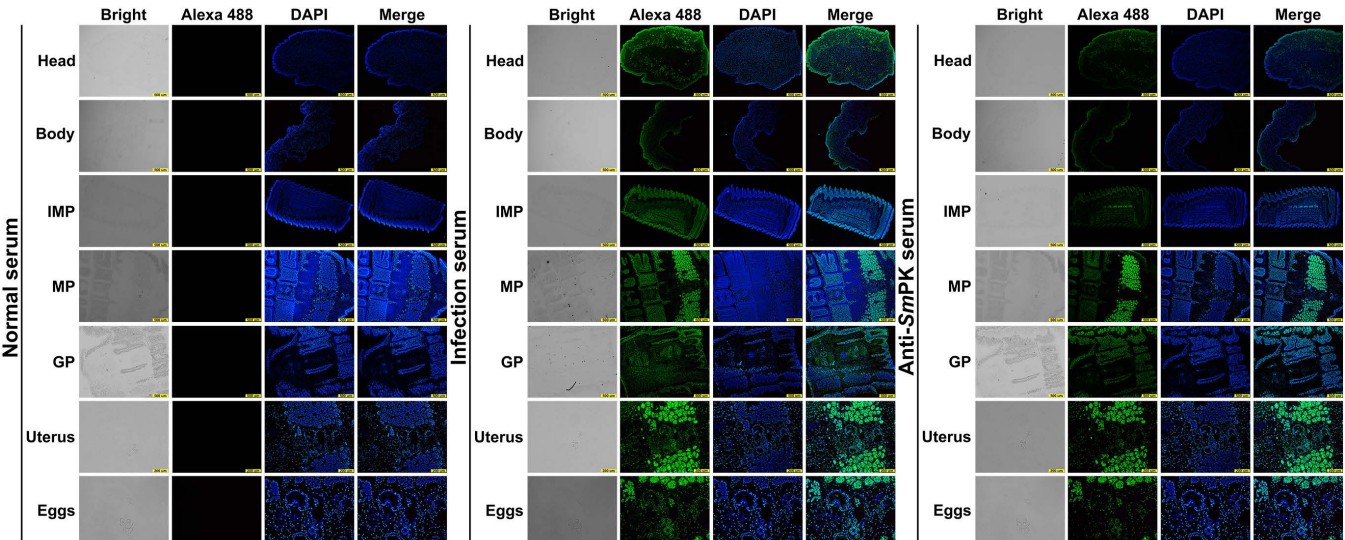

**Fig 9. Immunofluorescence analysis of PK1 in various life cycle stages of *Spirometra mansoni*.** Head: head of the plerocercoid; Body: body of the plerocercoid; IMP: immature proglottid; MP: mature proglottid; GP: gravid proglottid; Eggs: eggs in the uterus of the mature proglottid. **A.** Normal serum; **B.** Infected serum; **C.** Anti-rSmPK serum. GPR, scale of head, body, IMP, MP and GP: 500 μm; uterus: 200 μm; eggs: 100 μm.

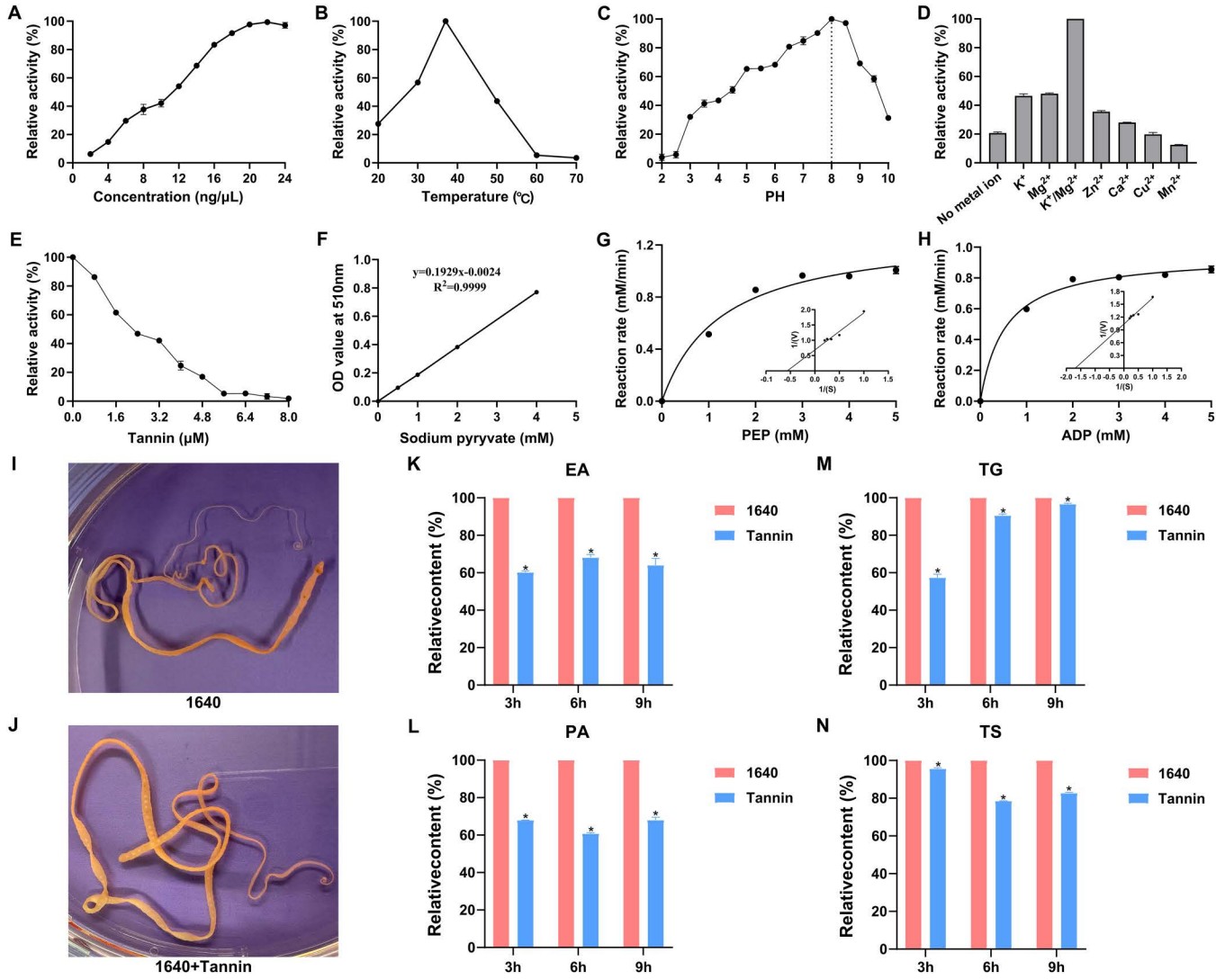

**Fig 10. Enzymatic activity analysis of rSmPK1 and evaluation of the metabolic inhibition effect of tannic acid on *S. mansoni*.** rSmPK1 was incubated with 2.5 mm PEP and 1.25 mm ADP for 10 min under various conditions. The optimal catalytic conditions for rSmPK1 were assessed with various rSmPK1 concentrations (2–24 ng/µL), temperatures (20–70 °C) and buffer solutions with different pH values (2–10). A. The optimum catalytic concentration of rSmPK1 is 20 ng/µl. B. The optimum catalytic temperature of rSmPK1 is 37 °C. C. The optimum catalytic pH of rSmPK1 is 8.0. D. Effects of different metal ions on rSmPK1 activity. $K^+$ and $Mg^{2+}$ clearly increase rSmPK1 activity. E. Inhibitory effects of tannic acid on rSmPK1 enzymatic activity. F. Standard curve of sodium pyruvate. pH 8.0 and 37 °C: Michaelis–Menten curves and Lineweaver–Burk plots of G. PEP and H. ADP. I. Negative control. J. Experimental group. K. Effect of tannic acid on the EA of *S. mansoni*. L. Effect of tannic acid on the PA content of *S. mansoni*. M. Effect of tannic acid on the TG content of *S. mansoni*. N. Effect of tannic acid on the TS content of *S. mansoni*. EA: enzyme activity; PA: pyruvic acid; TG: triglyceride; TS: total sugar; Statistical analysis was performed using SPSS 26 software. Statistical significance was determined by one-way analysis of variance (ANOVA), with data derived from three technical replicates. *$p$-value < 0.05 was considered to indicate statistical significance. The error bars represent the means ± standard deviations.

reduction of 3.4–42.65% ($p$ < 0.001), suggesting that the parasite preferentially decomposes triglycerides to rapidly generate ATP and meet urgent energy demands during the initial phase of metabolic disruption (Fig 10M). 4. The total sugar reserves were progressively depleted, decreasing by 4.34–21.46% ($p$ < 0.003), indicating that as triglyceride reserves were consumed, the parasite gradually broke down glycogen to maintain a stable energy supply (Fig 10N). In summary, the

inhibition of *Sm*PK1 activity affects the energy homeostasis of *S. mansoni* through a dual mechanism: initially, the parasite relies on rapid lipolysis (sharp consumption of triglycerides) as a compensatory pathway to sustain ATP production and meet immediate energy demands. However, over time, the energy supply shifts towards gradual consumption of glycogen reserves to meet long-term energy needs. This reprogramming of energy metabolism, potentially mediated by the regulation of related signalling pathways, may ultimately influence the differentiation process of *S. mansoni* proglottids.

## Discussion

All nine samples in this study presented relatively high-quality sequencing data (mean Q score = Q13, mapping rate > 96.95%), and the qPCR validation results were moderately correlated with the Nanopore sequencing results ($R^2$ = 0.6471), which is consistent with the results of Huang et al. [39]. Collectively, these results indicate that nanopore long-read sequencing generally provides more robust and established quantification for gene-level differential expression analysis. A total of 4,486 DEGs were identified among different segments of *S. mansoni.* Most of the DEGs were closely related to metabolic activities, which was consistent with previous research conclusions [17,18]. For different segments, the number of upregulated DEGs in MPs and GPs was greater than that in SNIPs, and the overall gene expression patterns in MPs and GPs were more similar to each other but significantly different from those in SNIPs. This result aligns with the findings of a study on *Moniezia expansa* [8] and *Hymenolepis microstoma* [40], suggesting that MPs and GPs adopt highly similar metabolic and developmental strategies during the segment development of cestodes.

To understand the functions of the DEGs, we first performed GO enrichment analysis. GO analysis of the coexpressed DEGs across the three proglottids revealed significant enrichment in glycolytic processes and metal ion binding. Glycolysis is a crucial pathway that allows organisms to rapidly obtain energy under anaerobic/hypoxic conditions. Its metabolic intermediates not only provide energy but also participate in macromolecular biosynthesis [21]. Metal ions play multiple important roles in organisms, including by acting as enzyme cofactors in catalysis, binding to macromolecules such as proteins to influence their functions, participating in redox reactions, and playing key regulatory roles in the immune system [41,42].

KEGG analysis of the DEGs revealed that glycolysis was a significantly enriched pathway, both for the DEGs coexpressed across the three proglottids and those between different proglottids. Glycolysis is crucial for energy metabolism in organisms. Xin et al. [43] demonstrated that pharmacological inhibition of glycolysis *in vitro* reduces *Echinococcus granulosus* protoscolex survival and induces ultrastructural damage. Corresponding *in vivo* studies have shown that glycolytic suppression significantly decreases *Echinococcus multilocularis* metacestode weight, disrupts metacestode integrity, and increases apoptosis, significantly impairing the growth and survival of metacestodes. Furthermore, integrated KEGG enrichment and GSEA revealed that glycolysis was more highly enriched in MPs and GPs than in SNIPs, with the highest activity observed in GPs. This may be closely related to the differentiation and development of *S. mansoni* proglottids. (1) As immature proglottids differentiate into MPs, more energy is required to support the development and maturation of reproductive organs. (2) As MPs further differentiate into GPs, the increasingly developed uterus and the development of eggs also demand more energy.

PPI analysis demonstrated that PK serves as a central node among all metabolism-related DEGs. PK catalyses the conversion of phosphoenolpyruvate (PEP) to pyruvate (PA), which is a key step in glycolysis that generates ATP, which is essential for cellular energy metabolism. Under aerobic conditions, pyruvate can also be converted into acetyl-CoA, which enters the citric acid cycle to participate in the metabolism of carbohydrates, fats, and proteins [19,22]. Previous studies have demonstrated that PK plays a vital role in the development, survival, and reproduction of parasites. Chen et al. [44] demonstrated that rat anti-recombinant *Clonorchis sinensis* PK (r*Cs*PyK) serum interferes with the energy metabolism of *C. sinensis*, significantly inhibiting its survival. Yue et al. [24] used RNA interference to silence the *Trichinella spiralis* PK (*Ts*PK) gene, which resulted in a reduction in *Ts*PK protein expression by 50.91% and in enzyme activity by 26.06%. However, the impact of PK on the differentiation and development of cestode segments has not been studied, and the

specific underlying mechanisms remain unclear. Thus, this study focused on PK for in-depth analysis, aiming to provide a theoretical foundation for elucidating the molecular biological mechanisms of *Sm*PKs and their role in the differentiation and development of *S. mansoni* segments.

All *Sm*PKs contain the conserved PK domain, indicating high conservation. Quantitative analysis of *Sm*PK expression at different life cycle stages revealed that PK1 and PK4 were highly expressed in MPs and GPs, suggesting that PK1 and PK4 may be more closely related to the segment development of *S. mansoni*. The number of PK family members varied among different species of medically important cestodes and trematodes, indicating the genetic diversity of PK genes. This dynamic variation in gene family size is closely linked to the adaptive evolution of species, potentially driven by gene rearrangements and transposition events that cause gene family expansion and contraction [45]. Phylogenetic analysis grouped the PK genes of Platyhelminthes into two major clades. Both cestodes and trematodes were present in Clades I and II, suggesting the existence of two distinct PK genotypes in these parasites. Interestingly, the PKs from the same group (e.g., the same family) tended to cluster together, indicating a certain degree of conservation of PKs within the same group.

Given its central role in PPI networks and high expression levels, *Sm*PK1 was selected as a representative member for cloning and functional characterization. ELISA and Western blot analysis demonstrated that r*Sm*PK1 exhibited good immunogenicity, and mouse immune serum specifically recognized the recombinant protein. The qPCR results revealed that *Sm*PK1 expression occurred across all life stages, with notably higher levels of MPs and GPs, likely due to their greater energy demands. The IFA results revealed that *Sm*PK1 was present mainly in the cortex of the plerocercoid, eggshell, and uterine and parenchymal tissues. This distribution pattern is consistent with the expression of PK in tissues with high energy metabolism, such as the cortex, uterus, and embryos. The regulation of complex life activities, gene regulation, and molecular modifications may influence protein localization, leading to differences in PK localization across different life stages [34,46]. Similar differences in PK localization have been observed in *Trichinella spiralis* and *Clonorchis sinensis* [24,44]. Furthermore, the localization of *Sm*PK1 closely aligns with that of other key energy metabolism enzymes, such as *Sm*LDH and *Sm*MDH [4,26], reflecting the energy metabolism requirements of *S. mansoni* at different life stages and highlighting the synergistic roles of these enzymes in maintaining energy balance.

In the buffer system, the metal ions $K^+$ and $Mg^{2+}$ significantly increased the enzyme activity of r*Sm*PK, suggesting that $K^+$ and $Mg^{2+}$ play indispensable roles in the activity of PK. $K^+$ and $Mg^{2+}$ have minimal effects on the overall structure of PK; however, they induce the reversal of specific domains, exposing the active site to a hydrophilic environment. This conformational change facilitates substrate binding and catalytic reactions, thereby increasing enzymatic activity [47]. Previous studies have demonstrated that tannic acid has the strongest inhibitory effect on pyruvate kinase, both in enzyme activity assays and *in vitro* growth inhibition experiments. Moreover, compared with other inhibitors, tannic acid is less cytotoxic [48]. Therefore, we selected tannic acid as the inhibitor to determine the enzyme activity. Tannic acid inhibited r*Sm*PK1 enzymatic activity in a concentration-dependent manner, resulting in complete inhibition at 8 μM. An et al. [48] reported that 20 μM tannic acid treatment inhibited *Babesia microti* growth by 80% after 3 d of incubation, and the inhibitory effect was further enhanced when the concentration was increased to 100 μM. Yue et al. [24] reported that enzymatic activity of native *Trichinella spiralis* PK in muscle larvae proteins was reduced by 12.82, 20.45, 51.77 and 65.65% respectively under various doses of tannic acid (25, 50, 75 and 100 μM) treatment. Li et al. [32] reported that *Babesia duncani* growth was inhibited by nearly 100% under 10 μM tannic acid treatment, suggesting a potential link between PK inhibition and growth suppression. We further investigated the role of *Sm*PK1 in the energy metabolism and development of *S. mansoni* by treating adult cestodes with tannic acid. *In vitro* treatment with 100 μM tannic acid for 3, 6, and 9 h reduced PK activity in proglottids by 39.78%, 31.87%, and 35.91%, respectively, demonstrating that tannic acid significantly inhibited *Sm*PK1 activity and reduced pyruvate production during glycolysis. This process triggers an energy metabolism crisis, forcing the cestode to compensate through rapid lipolysis and delayed glycogen depletion [24]. In this process, PK1 acts as a core regulator of metabolic reprogramming, playing a critical role in modulating the energy metabolism of the cestode.

Interestingly, transcriptomic analysis revealed the progressive upregulation of *Sm*PK1 expression from SNIPs to MPs and GPs, paralleling the increased activity of the glycolysis pathway. Moreover, previous studies have confirmed that during the intestinal parasitic stage, tapeworms rely primarily on glycolysis-based fermentative metabolism. This process generates end products such as acetate and propionate, enabling highly efficient energy production at approximately 5 ATP/glucose. Functionally, this metabolic mode resembles the malate dismutation pathway, representing an evolved anaerobic energy–generating strategy that allows tapeworms to adapt to hypoxic environments [49,50]. Although pyruvate kinase does not directly participate in this pathway, the pyruvate it produces during glycolysis serves as a fundamental substrate for the malate dismutation pathway. This study revealed that alterations in pyruvate kinase activity directly affect the production of pyruvate in glycolysis, thereby indirectly influencing the progression of the malate dismutation pathway. Consequently, these changes impact both the energy metabolism and survival of the tapeworm. Given the pivotal role of PK in glucose metabolism, we propose that *Sm*PK1 may regulate cestode segment differentiation and maturation by modulating the glycolysis pathway. Notably, as a natural polyphenolic compound, tannic acid possesses multiple protein–binding sites, suggesting the possibility of off–target effects [51]. However, some studies have revealed relatively specific mechanisms of action. For instance, in cancer research, Yang et al. [52] reported that tannic acid directly and selectively binds to the K433 site of PKM2, inhibiting its activity without affecting that of PKM1 and thereby inducing apoptosis in cancer cells. Nevertheless, given the structural differences between parasite and mammalian proteins, the potential off–target effects of tannic acid in parasites require further elucidation. Notably, this study preliminarily validated the functional role of pyruvate kinase in the growth and development of *Spirometra mansoni* through in vitro enzyme inhibition experiments. However, owing to the challenges in sample availability and the lack of a long–term in vitro culture system for the parasite, this study did not employ more advanced genetic techniques (such as RNA interference or CRISPR/Cas9 gene editing) to perform loss–of–function experiments for direct validation of gene function. Once a stable culture system is established in the future, further targeted genetic experiments will be conducted to provide more critical evidence for target validation.

## Conclusions

This study presents the first comprehensive characterization of gene expression across different proglottids of *S. mansoni*, identifying 4,486 DEGs. Comparative transcriptomic analysis revealed that PK expression levels gradually increased in SNIPs, MPs, and GPs as the segments developed and matured. Concurrently, the activity of the glycolysis pathway also progressively increased in SNIPs, MPs, and GPs, suggesting that PK may play a critical role in segment differentiation. Further analysis of the *Sm*PK family identified four *Sm*PK members (*Sm*PK1–*Sm*PK4). *Sm*PKs exhibit significant genetic diversity. The functional characterization of *Sm*PK1 demonstrated that both serum from immunized mice and serum from infected mice could specifically recognize r*Sm*PK1. The enzymatic activity of r*Sm*PK1 was found to be optimal at 37 °C and pH 8.0, with $K^+$ and $Mg^{2+}$ significantly influencing its activity. *In vitro* experiments indicated that tannic acid markedly inhibited *Sm*PK1 activity, reducing pyruvate production and inducing an energy metabolism crisis. Integrated omics analysis suggested that PK likely regulates segment differentiation and maturation by modulating the glycolysis pathway.

## Supporting information

**S1 Table. Sequence information of *cox*1 gene used for the identification of spargana isolates.**
(XLSX)

**S2 Table. Primer sequences for qRT–PCR validation.**
(XLSX)

**S3 Table. Primer sequences for qRT–PCR of *Sm*PKs.**
(XLSX))

**S4 Table.  Redundancy-reduced data statistics.**
(XLSX)

**S5 Table.  Statistics of differentially expressed genes between SNIPs and MPs.**
(XLSX)

**S6 Table.  Statistics of differentially expressed genes between MPs and GPs.**
(XLSX)

**S7 Table.  Statistics of differentially expressed genes between SNIPs and GPs.**
(XLSX)

**S8 Table.  Statistics for differentially coexpressed genes among different segments of *S. mansoni*.**
(XLSX)

**S9 Table.  Interactions among coexpressed metabolism-related genes among different segments of *S. mansoni*.**
(XLSX)

**S10 Table.  Interactions among metabolism-related genes between SNIPs and MPs.**
(XLSX)

**S11 Table.  Interactions among metabolism-related genes between MPs and GPs.**
(XLSX)

**S12 Table.  Interactions among metabolism-related genes between SNIPs and GPs.**
(XLSX)

**S13 Table.  PK sequences identified in medically important Cestoda and Trematoda.**
(XLSX)

**S1 Fig.  Principal component analysis of nanopore sequencing data across three types of proglottids.**
(PDF)

**S2 Fig.  Screening of differentially expressed genes.**
(PDF)

**S3 Fig.  GO analysis of DEGs between SNIPs and MPs.**
(PDF)

**S4 Fig.  GO analysis of DEGs between MPs and GPs.**
(PDF)

**S5 Fig.  GO analysis of DEGs between SNIPs and GPs.**
(PDF)

**S6 Fig.  KEGG analysis of DEGs between SNIPs and MPs.**
(PDF)

**S7 Fig.  KEGG analysis of DEGs between MPs and GPs.**
(PDF)

**S8 Fig.  KEGG analysis of DEGs between SNIPs and GPs.**
(PDF)

**S9 Fig. Analysis of the glycolysis pathway in MPs and GPs.**
(PDF)

**S10 Fig. Correlation of the expression levels of DEGs between the RNA-seq and qRT–PCR results.**
(PDF)

**S11 Fig. Characterization of the basic physicochemical properties of *Sm*PK1.**
(PDF)

## Author contributions

**Data curation:** Ke Zhou, Si Si Ru.

**Formal analysis:** Ke Zhou, Cheng Yue Cao, Rui Jie Wang, Jie Hao, Xi Zhang.

**Funding acquisition:** Xi Zhang.

**Methodology:** Ke Zhou, Cheng Yue Cao, Rui Jie Wang, Jie Hao, Xi Zhang.

**Project administration:** Xi Zhang.

**Resources:** Ke Zhou, Si Si Ru.

**Software:** Ke Zhou, Cheng Yue Cao, Si Si Ru, Xi Zhang.

**Supervision:** Xi Zhang.

**Validation:** Si Si Ru, Rui Jie Wang.

**Writing – original draft:** Ke Zhou.

**Writing – review & editing:** Xi Zhang.

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
