## [Decision Letter · Decision Letter 0]

1 Jul 2025

Transcriptomic analysis reveals that pyruvate kinase plays an important role in the differentiation of Spirometra mansoni proglottids by regulating the glycolysis/gluconeogenesis pathway

Dear Dr. Zhang,

Thank you for submitting your manuscript to PLOS Neglected Tropical Diseases. After careful consideration, we feel that it has merit but does not fully meet PLOS Neglected Tropical Diseases's publication criteria as it currently stands. Therefore, we invite you to submit a revised version of the manuscript that addresses the points raised during the review process.

Please submit your revised manuscript within 60 days Aug 30 2025 11:59PM. If you will need more time than this to complete your revisions, please reply to this message or contact the journal office at plosntds@plos.org. Please include the following items when submitting your revised manuscript:

We look forward to receiving your revised manuscript.

Kind regards,

Bruce A. Rosa

Academic Editor

Jong-Yil Chai

Section Editor

Shaden Kamhawi

co-Editor-in-Chief

Paul Brindley

co-Editor-in-Chief

**Additional Editor Comments :**

The authors should carefully consider the reviewer concerns, including sharing data indicating the data underlying the genotyping of S. mansoni.

As pointed out by reviewer 2, based on the link provided, it seems that the S. erinaceieuropaei genome reference was used for the analysis, but this is not acknowledged in the text except for in the link provided. If a different species or reference was used then this needs to be clarified, since it affects all downstream results. It is not appropriate to use a different species assembly to do a gene-level analysis since different species may have substantial expansions and contractions of gene families. Also as mentioned by reviewers, this is not a chromosomal-level assembly so the chromosome analysis in Figure 6 should be also be clarified or changed to scaffolds matching the assembly numbers. The mapping rates are also very high for a fragmented draft assembly to the wrong species, so it's unclear what was done. In a previous study (PMID:36406076), the submitting research group cited the S. erinaceieuropaei genome paper after saying that the S. mansoni genome has been published, so perhaps they consider these to be the same species or that the genome is incorrectly labeled?

Importantly, raw sequence data for each sample must be uploaded to a repository in order to comply with PLOS NTD Data availability requirements (https://journals.plos.org/plosntds/s/data-availability). Given the apparent confusion with much of the analysis, in order to make the data reproducible, supporting information tables with differential expression statistics for each gene in each comparison would also help to facilitate the usability of the data for other researchers.

Reviewer 1 provided a reference suggesting that gluconeogenesis is not present in tapeworms or other parasitic flatworms. I wasn’t able to access a copy of the book referenced to confirm this, but please perform some additional research to check on this. The text could be revised to just say glycolysis with no emphasis on gluconeogenesis, without any substantial changes.

Also, please ensure that statistical approaches are sufficiently described, including in figure captions.

**Journal Requirements:**

At this stage, the following Authors/Authors require contributions: Ke Zhou, Cheng Yue Cao, Si Si Ru, Rui Jie Wang, Jie Hao, and Xi Zhang. Please ensure that the full contributions of each author are acknowledged in the "Add/Edit/Remove Authors" section of our submission form.

2) Tables should not be uploaded as individual files. Please remove these files and include the Tables in your manuscript file as editable, cell-based objects. For more information about how to format tables, see our guidelines:

https://journals.plos.org/plosntds/s/tables 

3) Some material included in your submission may be copyrighted. According to PLOSu2019s copyright policy, authors who use figures or other material (e.g., graphics, clipart, maps) from another author or copyright holder must demonstrate or obtain permission to publish this material under the Creative Commons Attribution 4.0 International (CC BY 4.0) License used by PLOS journals. Please closely review the details of PLOSu2019s copyright requirements here: PLOS Licenses and Copyright. If you need to request permissions from a copyright holder, you may use PLOS's Copyright Content Permission form.

Potential Copyright Issues:

i) Please confirm (a) that you are the photographer of 1, 10I, and 10J, or (b) provide written permission from the photographer to publish the photo(s) under our CC BY 4.0 license.

ii) Figure 1. Please confirm whether you drew the images / clip-art within the figure panels by hand. If you did not draw the images, please provide (a) a link to the source of the images or icons and their license / terms of use; or (b) written permission from the copyright holder to publish the images or icons under our CC BY 4.0 license. Alternatively, you may replace the images with open source alternatives. See these open source resources you may use to replace images / clip-art:

4) Thank you for stating "The raw transcriptome data were submitted to the public database NCBI (Bioproject No. PRJNA1248218)." Please note that, though access restrictions are acceptable now, your entire minimal dataset will need to be made freely accessible if your manuscript is accepted for publication. 

5) Your current Financial Disclosure states, "The author(s) received no specific funding for this work. However, your funding information on the submission form indicates receiving funds. Please ensure that the funders and grant numbers match between the Financial Disclosure field and the Funding Information tab in your submission form. Note that the funders must be provided in the same order in both places as well.

Please amend your detailed Financial Disclosure statement. This is published with the article. It must therefore be completed in full sentences and contain the exact wording you wish to be published.

1) Please clarify all sources of financial support for your study. List the grants, grant numbers, and organizations that funded your study, including funding received from your institution. Please note that suppliers of material support, including research materials, should be recognized in the Acknowledgements section rather than in the Financial Disclosure

2) State the initials, alongside each funding source, of each author to receive each grant. For example: "This work was supported by the National Institutes of Health (####### to AM; ###### to CJ) and the National Science Foundation (###### to AM)."

3) State what role the funders took in the study. If the funders had no role in your study, please state: "The funders had no role in study design, data collection and analysis, decision to publish, or preparation of the manuscript."

4) If any authors received a salary from any of your funders, please state which authors and which funders..

**Reviewers' Comments:**

Reviewer's Responses to Questions

**Key Review Criteria Required for Acceptance?**

**Methods**

-Are the objectives of the study clearly articulated with a clear testable hypothesis stated?

-Is the study design appropriate to address the stated objectives?

-Is the population clearly described and appropriate for the hypothesis being tested?

-Is the sample size sufficient to ensure adequate power to address the hypothesis being tested?

-Were correct statistical analysis used to support conclusions?

-Are there concerns about ethical or regulatory requirements being met?

Reviewer #1: -Are the objectives of the study clearly articulated with a clear testable hypothesis stated? Not entirely.

-Is the study design appropriate to address the stated objectives? Not entirely.

-Is the population clearly described and appropriate for the hypothesis being tested? Not in all the experiments.

-Is the sample size sufficient to ensure adequate power to address the hypothesis being tested? Not in all the experiments.

-Were correct statistical analysis used to support conclusions? Not in all the experiments.

-Are there concerns about ethical or regulatory requirements being met? No.

Reviewer #2: -Are the objectives of the study clearly articulated with a clear testable hypothesis stated?

YES

The general objectives of the study are described in the introduction, particularly the focus on proglottid differentiation in Spirometra mansoni and the role of pyruvate kinase in this process. However, the manuscript does not explicitly present a clear, testable hypothesis in a formal manner. The research is guided more by exploratory transcriptomic comparison than by a defined experimental hypothesis

-Is the study design appropriate to address the stated objectives?

YES

Overall, the study design is appropriate for addressing the stated objectives. The authors employ full-length Nanopore transcriptome sequencing across three types of proglottids of Spirometra mansoni, followed by differential gene expression analysis, functional enrichment, and characterization of pyruvate kinase genes. The combination of comparative transcriptomics with molecular and enzymatic assays supports the investigation of gene function in proglottid differentiation.

-Is the population clearly described and appropriate for the hypothesis being tested?

NO

While the authors state that the parasite was identified as S. mansoni using the genotyping method of Kuchta et al. 2021, they do not present any results from this analysis—such as the gene marker used, GenBank accession numbers, phylogenetic placement, or sequence similarity.

-Is the sample size sufficient to ensure adequate power to address the hypothesis being tested?

YES

The sample size (n = 3 biological replicates per proglottid type) is minimal but acceptable for exploratory transcriptomic studies.

-Were correct statistical analysis used to support conclusions?

YES

the statistical tools used (DESeq2, GOseq, KOBAS, GSEA, STRING) are appropriate and standard for transcriptomic data analysis.

-Are there concerns about ethical or regulatory requirements being met?

NO

Ethical approval is clearly stated, including compliance with national guidelines and approval by the Life Science Ethics Committee of Zhengzhou University (Approval No. ZZUIRB GZR 2022-0062).

Reviewer #3: (No Response)

**Results**

-Does the analysis presented match the analysis plan?

-Are the results clearly and completely presented?

-Are the figures (Tables, Images) of sufficient quality for clarity?

Reviewer #1: Does the analysis presented match the analysis plan?

Yes, the analyses are consistent with the stated analysis plan.

Are the results clearly and completely presented?

Yes, the results are presented clearly and comprehensively.

Are the figures (Tables, Images) of sufficient quality for clarity?

Yes, the figures and tables are of sufficient quality and support clarity of interpretation.

Reviewer #2: -Does the analysis presented match the analysis plan?

YES

the described analyses—differential expression using DESeq2, enrichment analyses (GO, KEGG, GSEA), and PPI network construction—are consistent with the study’s stated objectives and methodology. However, some analytical choices, such as focusing primarily on coexpressed DEGs, are not fully justified in the plan and would benefit from clarification.

-Are the results clearly and completely presented?

NO

The main results are quite clear and follow a logical structure. However, some areas lack detail or supporting data—for example, species identification of S. mansoni is stated but not shown. The manuscript repeatedly refers to the use of the S. mansoni genome, which has not been sequenced; only in one instance is an accession number provided, which actually corresponds to S. erinaceieuropaei, indicating inconsistency and a need for clarification. Also DEG numbers sometimes differ between text and figures. The rationale for focusing on coexpressed DEGs is also not fully explained. The manuscript does not state where the sequencing data will be made available. To ensure transparency and reproducibility, the authors should clearly indicate the repository (e.g., NCBI SRA, ENA, or CNGBdb) where the raw transcriptomic data will be deposited, along with relevant accession numbers.

Are the figures (Tables, Images) of sufficient quality for clarity?

no

while the overall quality and resolution of figures is acceptable, Figure 2A raises concerns. Specifically, some bars representing DEGs are taller than those labeled “All,” which is conceptually inconsistent, as DEGs should form a subset of all annotated genes. This suggests possible mislabeling or unclear grouping of data. The figure would benefit from clearer explanation in the legend or main text to avoid misinterpretation.

Figure 6A refers to “chromosomal localization” of SmPKs. However, the authors used the fragmented draft genome of Spirometra erinaceieuropaei, which is not at chromosome level. Therefore, this panel is misleading. The term “scaffold-level localization” would be more accurate, or clarification in the figure legend and methods is needed to avoid overstating the genome assembly quality.

The figure legends in the manuscript are generally overly brief and lack sufficient detail. As they currently stand, the figures cannot be fully understood without referring to the main text, which limits their standalone interpretability. This is particularly problematic for readers who rely on figure legends to quickly grasp experimental results.

For example, the legend for Figure 3B simply states: “B. GSEA of the glycolysis/gluconeogenesis pathway.” This provides no information on:

what comparison was made (e.g., which proglottid stages were analyzed),

what the enrichment score represents,

or how the result should be interpreted in a biological context.

More informative and self-contained figure legends are strongly recommended.

Reviewer #3: The results are clearly presented and the figures are generally of sufficient quality. However, some supplementary analysis could help increase the overall reliability of the differential expression results.

1. it would be helpful to provide results such as PCA plots, hierarchical clustering heatmaps, or pairwise correlation analyses to demonstrate the consistency and expected separation among the biological replicates. Including this information would improve the transparency of the data quality control and can help increase the overall reliability of the differential expression results.

2. I would also encourage the authors to check whether their Nanopore long-read data enabled the identification of any novel isoforms or previously unannotated gene structures. Presenting such findings, even briefly as supplementary information, would further demonstrate the added value of using Nanopore full-length sequencing and would make the dataset more useful for future studies on S. mansoni transcriptome complexity.

3. For Figure 8B, please include statistical significance testing for the qPCR results to confirm that the observed differences are robust.

**Conclusions**

-Are the conclusions supported by the data presented?

-Are the limitations of analysis clearly described?

-Do the authors discuss how these data can be helpful to advance our understanding of the topic under study?

-Is public health relevance addressed?

Reviewer #1: -Are the conclusions supported by the data presented?

Yes, but the conclusions are incomplete.

-Are the limitations of analysis clearly described?

No, they are not clearly described.

-Do the authors discuss how these data can be helpful to advance our understanding of the topic under study?

It is discussed briefly, but the discussion is far from complete for the field.

-Is public health relevance addressed?

The issue is not fully addressed.

Reviewer #2: Are the conclusions supported by the data presented?

Yes

with some reservations. The main conclusions—that pyruvate kinase (especially SmPK1) plays a central role in energy metabolism and potentially in the differentiation of S. mansoni proglottids—are generally supported by transcriptomic data, functional assays, and inhibition experiments. However, some claims (e.g., about direct involvement of PK in differentiation) remain somewhat speculative and would benefit from stronger causal evidence (e.g., knockdown studies).

Are the limitations of analysis clearly described?

NO

The manuscript does not adequately discuss the limitations of the analysis. Important aspects such as the use of a genome from S. erinaceieuropaei instead of S. mansoni, the lack of independent validation of species identity, absence of functional knockdown experiments to confirm gene function, and potential limitations of in vitro assays are not addressed. A more transparent discussion of these issues would strengthen the credibility and interpretation of the results.

Is public health relevance addressed?

NO Although Spirometra mansoni is a zoonotic parasite with public health importance, the manuscript does not discuss the relevance of the findings in the context of human infection or disease.

Reviewer #3: 1. The study demonstrates, through comparative transcriptomic analysis, enzyme activity assays, and inhibitor experiments, that PKs play a key role in the energy metabolism of tapeworms. However, it would be helpful for the authors to explicitly acknowledge that direct experimental evidence linking PKs to the regulation of differentiation through glycolysis is still lacking and that this remains a potential area for future study. I therefore recommend that the authors refine the title to more precisely reflect the scope of the study’s findings.

**Editorial and Data Presentation Modifications?**

Reviewer #1: The manuscript would benefit from language editing in multiple sections, as shown by examples provided in the Summary and General Comments.

Reviewer #2: Abstract:

Please specify which types of proglottids were analyzed.

Several sentences in the abstract and author summary are identical. This redundancy should be eliminated.

Redundant example:

"Phylogenetic analysis revealed that SmPKs have undergone varying degrees of evolution and exhibit high diversity. The optimal reaction conditions for recombinant SmPK1 (rSmPK1) were 37°C and pH 8.0, and the addition of K+/Mg2+ significantly enhanced its catalytic activity. Tannic acid could significantly inhibit the activity of SmPK1 in vitro..."

This is repeated verbatim in both sections with only minor terminological changes. Please revise to avoid duplication.

Introduction:

The introduction should mention pyruvate kinase (PK) and its importance in parasite metabolism, given its central role in the study.

Latin binomials (e.g., Spirometra mansoni) should be spelled in full on first mention (lines 95-96).

The section on nanopore sequencing (lines 102–114) is tangential and could be condensed or removed.

The sentence identifying the proglottid types (lines 116-117) should be incorporated into the abstract.

Parasites and Animals:

The species identification via Kuchta et al.'s method is stated but no supporting data are provided. Include:

Gene marker used

GenBank accession numbers

Sequence similarity or phylogenetic placement

At minimum, include this data in a supplementary table

Materials and Methods:

Line 145: CoWin Biosciences is a distributor; the TRIzol manufacturer is Invitrogen.

Lines 152–155: Missing critical information about software tools for quality filtering, rRNA removal, primer detection, and FLNC identification. Please add.

Lines 161–165: Clarify how CPM was used. DESeq2 does not use CPM for normalization.

Lines 168–176 (Annotation and Enrichment):

Specify database versions or download dates.

Justify use of both COG and KOG.

Name the GSEA software/tool used.

Include STRING database version and confidence threshold.

Provide Cytoscape version.

Lines 194–202:

The discussion of PK function belongs in the Introduction. The Methods should focus on identification criteria for SmPKs.

Lines 211–213:

Clarify genome source. Mapping was done against S. erinaceieuropaei (line 157), not S. mansoni. Unify terminology and remove misleading statements.

Lines 213–216:

Provide software version for TBtools2.

qRT-PCR Section:

Include summarized protocol (referencing Liu et al. is fine).

Clarify whether GAPDH stability was verified.

State whether technical replicates were used.

Line 226:

Use "medically important cestodes" instead of "medical cestodes".

Line 285:

The gene count (26,270) seems inflated. Clarify if this includes isoforms, all transcripts, or coding genes. Define criteria for gene prediction.

Line 286:

Define "novel genes" and the method of identification (e.g., BLAST, absence in reference genome).

GO and KEGG Analysis:

Focus only on co-expressed DEGs may overlook stage-specific signals. Include condition-specific analyses.

Figure 2A:

The DEGs bar height exceeding the "All" bar is misleading. Clarify calculation or replot with enrichment (e.g., fold change or p-value).

Venn Diagram vs DEG Counts:

Numbers in Venn diagrams (Fig. S1a) differ from those in text. If only annotated genes are shown, clarify this.

Same applies for KEGG: counts in text differ from Fig. 2a. Clarify filtering criteria and ensure consistency.

Species Identification:

Despite claims, the S. mansoni genome does not exist. The provided accession number matches S. erinaceieuropaei. This must be corrected throughout.

Reviewer #3: 1. the temperature unit in line 71 is formatted correctly by adding a space between the numeric value and the degree Celsius symbol (e.g., “37 °C” instead of “37°C”). In addition, the same issue exits in the Method section.

2. I suggest that the authors expand the Introduction to include a brief overview of relevant previous studies that have used omics approaches to investigate the molecular mechanisms of differentiation in sparganosis This additional context would help highlight the knowledge gap addressed by the present study and emphasize the novelty and significance of applying full-length transcriptomic analysis to S. mansoni.

**Summary and General Comments**

Reviewer #1: Overview

The manuscript by Zhou and collaborators "Transcriptomic analysis reveals that pyruvate kinase plays an important role in the differentiation of Spirometra mansoni proglottids by regulating the glycolysis/gluconeogenesis pathway"presents an interesting work that studies the differentiation of proglottids in the cestode Spirometra mansoni by transcriptomics. While the study is clearly presented and the figures and tables are well prepared, it lacks a deep biological interpretation of features relevant to cestode physiology.

Below, I provide my critical assessment of the manuscript.

Gluconeogenesis is not present in tapeworm and other parasitic flatworms (A.G.M. Tielens, J.J. van Hellemond, Unusual aspects of metabolism in flatworm parasites, 2006). This is critical for what the authors show in the work and also in the title. This is an error probably produced by the analysis of the GO terms obtained, these are powerful tools but not always directly useful for tapeworms and other non-classic organisms, where a deeper biological analysis is needed to interpret the results.

The final experiment shown in the text is not properly set up. - Lines 262-271 - In vitro” inhibition assay: “Two adult cestodes with similar individual characteristics…” Based on the experimental description, it is difficult to draw reliable conclusions from this part of the study.

These issues, together with the other points I mention below indicate that this work is not ready for publication. I recommend re-analysing the data, taking into account the particular metabolic characteristics already described in cestodes, and comparing the results with previous transcriptomic studies that analysed adult tapeworms.

Given that these key biological considerations are not addressed, the manuscript requires substantial revision before it can be considered for publication.

Other points

- "medical platyhelminths" is an awkward and non-standard expression, it appears several times in text.

- Fig 9. an explanation of why some of the tissues shown are almost identical is required. Example: “Infection serum, Uterus, Alexa 488” and “Anti-SmPK serum, Uterus, Alexa 488” and some others are almost identical, or very similar in structure, why is this the case?

- There are other terms that are awkward in the text, like in Line 431 “significant fluorescence”.

Reviewer #2: This manuscript explores the role of pyruvate kinase in the segmental differentiation of Spirometra mansoni through full-length transcriptomic profiling and complementary biochemical analyses. The topic is timely and relevant, especially given the limited understanding of energy metabolism in cestode development. The use of Nanopore sequencing alongside functional assays provides a potentially valuable dataset and offers insights into metabolic regulation during proglottid maturation.

However, the overall execution of the study falls short in several key areas. Methodological descriptions are often lacking in detail, especially concerning the source of genomic data, parasite species confirmation, and analysis reproducibility. Critical background information (such as the identity and validation of the species used) is not adequately documented, which undermines confidence in the conclusions. Moreover, the interpretation of data sometimes overreaches the presented evidence, with speculative links made between metabolic changes and differentiation in the absence of causal functional validation.

The manuscript would also benefit from clearer and more informative figure legends, consistent use of terminology, and removal of duplicated or overly general text across sections. Some figures are misleading in how data are labeled or described, and others lack standalone clarity. In particular, the reliance on a genome from S. erinaceieuropaei—without consistent disclosure or justification—requires correction and clarification.

Despite these concerns, the study has potential to make a meaningful contribution to the field of parasite developmental biology. If the authors can substantially revise the manuscript—providing missing validation data, improving clarity and figure presentation, and clearly acknowledging limitations—the work may become suitable for publication. As it stands, a major revision is necessary to address both technical and interpretative issues.

Reviewer #3: This manuscript presents a well-conducted and novel study investigating the molecular mechanisms underlying proglottid differentiation in Spirometra mansoni. The authors generated full-length transcriptomic data for three biologically distinct proglottid segments (scolex-neck-immature proglottid, mature proglottid, and gravid proglottid) using Nanopore sequencing technology. They identify pyruvate kinase (PK) as a central regulator involved in the glycolysis/gluconeogenesis pathway, validate its function through cloning and enzyme assays, and provide a comprehensive characterization of the SmPK gene family. The results are interesting and contribute valuable insights into the metabolic regulation of tapeworm differentiation.

Overall, the manuscript is clearly written and the experimental design is appropriate for the stated objectives. The data are presented in a logical and convincing manner, and the main conclusions are well supported by the evidence provided. I have only a few minor suggestions for improvement:

1. I recommend that the authors provide a brief description in the Introduction section explaining the biological differences among the three segments used for sequencing (scolex-neck-immature proglottid, mature proglottid, and gravid proglottid). This background would help readers better understand the biological context and developmental progression of Spirometra mansoni proglottids and the rationale for selecting these segments for comparative transcriptomic analysis. Adding this context will help readers unfamiliar with tapeworm biology better understand the link between segmental morphology and the regulation of differentiation.

2. While the Introduction clearly highlights the general advantages of Nanopore long-read sequencing, I recommend that the authors briefly acknowledge that Illumina short-read sequencing generally provides more robust and established quantification for gene-level differential expression analysis in Discussion. Additionally, since the authors have already generated Nanopore data, they could consider better leveraging the unique strengths of the long-read approach by including an isoform-level analysis or using the full-length reads to refine the gene models and improve the genome annotation. Adding such information, even at a summary level, would enhance the impact of the dataset and showcase the full advantage of combining expression quantification with structural transcriptome insights.

PLOS authors have the option to publish the peer review history of their article (what does this mean? ). If published, this will include your full peer review and any attached files.

**Do you want your identity to be public for this peer review?** For information about this choice, including consent withdrawal, please see our Privacy Policy .

Reviewer #1: No

Reviewer #2: **Yes: ** Roman Loentovyč

Reviewer #3: No

**Figure resubmission:**

**Reproducibility:**



---

## [Decision Letter · Decision Letter 1]

20 Aug 2025

Transcriptomic analysis reveals that pyruvate kinase potentially plays a key role in the differentiation of Spirometra mansoni proglottids by regulating the glycolysis pathway

Dear Dr. Zhang,

Thank you for submitting your manuscript to PLOS Neglected Tropical Diseases. After careful consideration, we feel that it has merit but does not fully meet PLOS Neglected Tropical Diseases's publication criteria as it currently stands. Therefore, we invite you to submit a revised version of the manuscript that addresses the points raised during the review process.

Please submit your revised manuscript within 60 days Sep 19 2025 11:59PM. If you will need more time than this to complete your revisions, please reply to this message or contact the journal office at plosntds@plos.org. Please include the following items when submitting your revised manuscript:

We look forward to receiving your revised manuscript.

Kind regards,

Bruce A. Rosa

Academic Editor

Jong-Yil Chai

Section Editor

Shaden Kamhawi

co-Editor-in-Chief

Paul Brindley

co-Editor-in-Chief

**Additional Editor Comments:**

Reviewer 1 has some additional suggestions for the manuscript which require an additional round of revision. Please address these carefully, and including additional literature support as described, and providing some additional justifications of inhibitor selections and concentrations, etc. For the in vitro data, the methods describe two replicates, so please be sure to mention the replicate counts and the specific statistical test used in the Fig 10 caption, and add error bars indicating the variability.

**Journal Requirements:**

**Reviewers' Comments:**

Reviewer's Responses to Questions

**Key Review Criteria Required for Acceptance?**

**Methods**

-Are the objectives of the study clearly articulated with a clear testable hypothesis stated?

-Is the study design appropriate to address the stated objectives?

-Is the population clearly described and appropriate for the hypothesis being tested?

-Is the sample size sufficient to ensure adequate power to address the hypothesis being tested?

-Were correct statistical analysis used to support conclusions?

-Are there concerns about ethical or regulatory requirements being met?

Reviewer #1: -Are the objectives of the study clearly articulated with a clear testable hypothesis stated?

Partially

-Is the study design appropriate to address the stated objectives?

Partially

-Is the population clearly described and appropriate for the hypothesis being tested?

Not completely

-Is the sample size sufficient to ensure adequate power to address the hypothesis being tested?

Not completely

-Were correct statistical analysis used to support conclusions?

Not completely

-Are there concerns about ethical or regulatory requirements being met?

In general, yes.

Reviewer #2: (No Response)

**Results**

-Does the analysis presented match the analysis plan?

-Are the results clearly and completely presented?

-Are the figures (Tables, Images) of sufficient quality for clarity?

Reviewer #1: Results

-Does the analysis presented match the analysis plan?

Yes.

-Are the results clearly and completely presented?

Yes.

-Are the figures (Tables, Images) of sufficient quality for clarity?

Partially

Reviewer #2: (No Response)

**Conclusions**

-Are the conclusions supported by the data presented?

-Are the limitations of analysis clearly described?

-Do the authors discuss how these data can be helpful to advance our understanding of the topic under study?

-Is public health relevance addressed?

Reviewer #1: -Are the conclusions supported by the data presented?

Not completely

-Are the limitations of analysis clearly described?

Not completely

-Do the authors discuss how these data can be helpful to advance our understanding of the topic under study?

Not completely

-Is public health relevance addressed?

Not completely

Reviewer #2: (No Response)

**Editorial and Data Presentation Modifications?**

Reviewer #1: (No Response)

Reviewer #2: (No Response)

**Summary and General Comments**

Reviewer #1: Transcriptomic analysis reveals that pyruvate kinase plays an important role in the differentiation of Spirometra mansoni proglottids by regulating the glycolysis/gluconeogenesis pathway

Overview

The work presented here by Zhou and collaborators shows a study of the differentiation of proglottids in the cestode Spirometra mansoni by transcriptomics. While the study is improved from the last version they presented, it still lacks a deep biological interpretation of features relevant to cestode biology. Below, I provide my critical assessment of the manuscript.

Major points

The authors focus on the metabolism of cestodes. However, they do not mention that cestodes possess a unique metabolic system. In particular, the malate dismutation pathway is not addressed in the results. Consequently, this significant aspect of cestode metabolism is not analyzed in the work.

Additionally, there are few comparisons of the results with other studies that have applied transcriptomics to adult cestodes. Notably, it is surprising that the authors do not compare their findings with RNA-seq results from different regions of adult Hymenolepis microstoma (Olson et al., 2018, https://doi.org/10.1186/s13227-018-0110-5).

The use of tannic acid as an inhibitor for the enzyme activity assay is not justified enough. It is not shown that the inhibition of tannic acid is specific. Why did the authors use this inhibitor and not others? Also, the authors do not discuss possible off-target effects of this compound in the results or discussion.

About the “In vitro inhibition assay”:

As described, the “In vitro inhibition assay” section lacks sufficient biological validity to support the conclusions drawn. The data appear to be based on only two individual worms; one treated with 100 µM tannic acid and one untreated control. This very high inhibitor concentration is not justified in the text, nor is it discussed in the limitations section.

No biological replicates are described, making it impossible to assess reproducibility or variability. Without independent replicates, these results should be considered preliminary. The authors should repeat the experiment with an appropriate number of biological replicates and controls before drawing definitive conclusions.

Furthermore, it is unclear how statistical analyses and error bars were generated from the experiment as described. The manuscript should clearly state the number of biological and technical replicates, the statistical tests used, and the nature of the error bars. Without this information, the statistical validity of the findings cannot be evaluated.

Other points

Figure 1 – As I mentioned before “Medical Tapeworm” is not an appropriate sentence, and it is here as a title in the image.

Line 160 – “Adult worms were obtained as previously described [26].” Please provide at least a short description of how this was carried out.

Line 163-164 – “Female Kunming mice, aged 4-6 weeks, were purchased from the Henan Laboratory Animal Center.” It is not in context why this is mentioned here.

Line 168 – It is not clear if the “samples = 3” for SNIP, MP and GP, are from three different worms or are 3 different sections of the same worms divided in three.

Figure 3 Part A – It is not clear what are the numbers in the figure.

Figure 3 Part C – The format of the font is not uniform in the image.

Reviewer #2: (No Response)

PLOS authors have the option to publish the peer review history of their article (what does this mean? ). If published, this will include your full peer review and any attached files.

**Do you want your identity to be public for this peer review?** For information about this choice, including consent withdrawal, please see our Privacy Policy .

Reviewer #1: No

Reviewer #2: **Yes: ** Roman Leontovyč

**Figure resubmission:**
---

## [Editor Report · Decision Letter 2]

16 Sep 2025

Dear Ph.D. Zhang,

We are pleased to inform you that your manuscript 'Transcriptomic analysis reveals that pyruvate kinase potentially plays a key role in the differentiation of Spirometra mansoni  proglottids by regulating the glycolysis pathway' has been provisionally accepted for publication in PLOS Neglected Tropical Diseases.

Best regards,

Bruce A. Rosa

Academic Editor

Jong-Yil Chai

Section Editor

Shaden Kamhawi

co-Editor-in-Chief

Paul Brindley

co-Editor-in-Chief

We are pleased to inform you that your second revised manuscript has been accepted for publication. We appreciate your kind cooperation.

---

## [Editor Report · Acceptance letter]

Dear Ph.D. Zhang,

We are delighted to inform you that your manuscript, "Transcriptomic analysis reveals that pyruvate kinase potentially plays a key role in the differentiation of Spirometra mansoni  proglottids by regulating the glycolysis pathway," has been formally accepted for publication in PLOS Neglected Tropical Diseases.

Best regards,

Shaden Kamhawi

co-Editor-in-Chief

Paul Brindley

co-Editor-in-Chief
